# IndEgo: A Dataset of Industrial Scenarios and Collaborative Work for Egocentric Assistants

**Vivek Chavan**[1,2]*    **Yasmina Imgrund**[2]†    **Tung Dao**[2]†    **Sanwantri Bai**[3]†
**Bosong Wang**[4]†    **Ze Lu**[5]†    **Oliver Heimann**[1]    **Jörg Krüger**[1,2]

[1]Fraunhofer IPK, Berlin    [2]Technical University of Berlin    [3]University of Tübingen
[4]RWTH Aachen University    [5]Leibniz University Hannover

Project Page: https://indego-dataset.github.io/

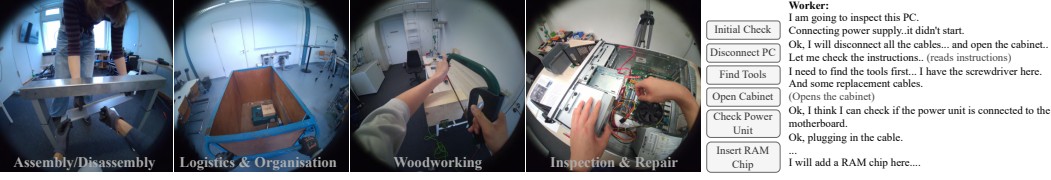

Figure 1: Some examples from the IndEgo dataset showing different industrial cases. **a.** Assembly/Disassembly and Collaborative Work (further elaboration on Figure 2), **b.** Logistics and Organisation , **c.** Woodworking , **d.** Inspection and Repair (The worker's narration and the annotated actions are also shown).

## Abstract

We introduce IndEgo, a multimodal egocentric and exocentric dataset addressing common industrial tasks, including assembly/disassembly, logistics and organisation, inspection and repair, woodworking, and others. The dataset contains 3,460 egocentric recordings (approximately 197 hours), along with 1,092 exocentric recordings (approximately 97 hours). A key focus of the dataset is collaborative work, where two workers jointly perform cognitively and physically intensive tasks. The egocentric recordings include rich multimodal data and added context via eye gaze, narration, sound, motion, and others. We provide detailed annotations (actions, summaries, mistake annotations, narrations), metadata, processed outputs (eye gaze, hand pose, semi-dense point cloud), and benchmarks on procedural and non-procedural task understanding, Mistake Detection, and reasoning-based Question Answering. Baseline evaluations for Mistake Detection, Question Answering and collaborative task understanding show that the dataset presents a challenge for the state-of-the-art multimodal models. Our dataset is available at: https://huggingface.co/datasets/FraunhoferIPK/IndEgo

## 1 Introduction

Egocentric Vision and AI are gaining significant attention, driven by both their economic potential and the recent development of general-purpose foundational models [1, 2, 3, 4]. One of the key goals

---

*Project Lead. Correspondence: contact@vivekchavan.com
†Work done during student theses/projects at Fraunhofer IPK, Berlin.

39th Conference on Neural Information Processing Systems (NeurIPS 2025) Track on Datasets and Benchmarks.

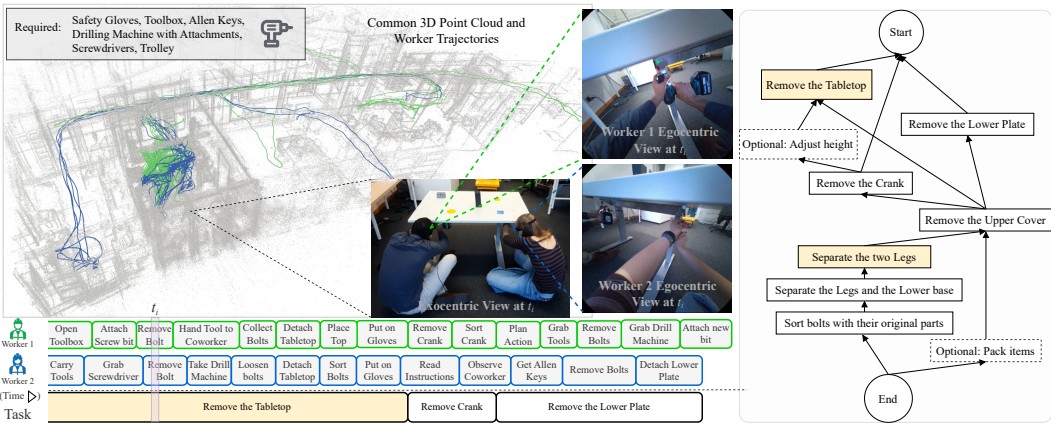

Figure 2: A scenario of a *disassembly* process from the IndEgo dataset. The two participants work collaboratively on the task. The semi-dense point cloud and the user trajectories are generated by processing the raw data from the Aria device [4, 11]. The egocentric perspective of the two participants with the projected eye gaze point can be seen in relation to the 3D environment and the exocentric view. **Bottom:** The annotations from each worker's perspective, and the keysteps in the process. **Right:** The corresponding task graph for the procedure. The flow of activities is from top to bottom, and dependencies are shown with an arrow. ▢ denotes labour-intensive steps.

of this field is to develop helpful AI assistants that understand user's actions, intentions, and needs, and provide valuable guidance [5, 4]. Such assistants would be valuable in learning and acquiring new skills, navigating new environments, and improving user experience. Additionally, egocentric research is also relevant to the development of embodied agents that can learn from demonstrations, assist and collaborate with humans [6, 7, 8, 9, 10].

To enable and accelerate research in these areas, several datasets and benchmarks have been introduced [12, 13, 3]. Current datasets and initiatives focus heavily on daily activities and procedures [12, 13, 14]. This is in part because these are applicable across different cultures and lifestyles, and also because it is relatively easier to design experiments and collect data in settings such as the kitchen [13, 15, 16]. Some recent datasets also focus on industry-like contexts [17, 18], however, the datasets published in true industrial settings remain low. Industrial scenarios offer a rich domain for egocentric vision research due to their complexity, diversity of tasks, and dynamic environments [18, 17]. Workers in industrial settings perform intricate manual operations, manipulate various tools and components, and navigate in cluttered and unpredictable spaces [19, 20]. These conditions present unique challenges for egocentric vision systems, such as accurately recognising actions and gestures in real-time, identifying and localising tools and parts amidst visual occlusions, and adapting to varying lighting and environmental conditions. These scenarios are underrepresented in current egocentric vision datasets and research.

Egocentric Vision-based assistive technologies can offer several unique and practical implications for improving productivity, safety, and efficiency in industrial operations. To investigate the future application potential of wearable egocentric vision-based assistants, we present the **IndEgo** dataset, consisting of diverse Industrial tasks and processes from an Egocentric perspective, with a supplementary Exocentric view for reference in several cases. IndEgo comprises 3,460 unique egocentric video recordings (totalling 197.1 hours), and 1,092 (totalling 96.8 hours) accompanying exocentric video recordings. The dataset includes scripted and unscripted actions from participants in a dedicated industrial facility, collected in diverse conditions (lighting, time of the day, setup). Figure 1 shows an example. The dataset also includes collaborative work, where two participants work together on a common task in various roles (partners, teacher-student, leader-assistant). AI assistants and embodied agents of the near future will work alongside other humans on such tasks, which makes this a relevant subdomain for further exploration and research.

We recommend checking the arXiv version of this paper for minor updates or clarifications: https://arxiv.org/abs/2511.19684

## 2 Related Work

**AI Assistants.** The mainstream adoption of AI technology has led to a strong focus on the development of digital assistants that can chat (text and audio), answer questions, understand images, videos, and process multimodal data [21, 2, 22]. Research on embodied agents that can manipulate their environment and assist humans is also accelerating [23, 24, 25, 26].

**Egocentric Computer Vision.** The goal is to understand the world from a human-centric perspective [27, 3, 28]. Users often collect data by wearing a portable camera device (or other sensors) and performing various activities [12, 13]. This provides insight into several scenarios that cannot be captured by traditional vision research [12, 5]. This introduces several novel challenges, due to the dynamic nature of the scenes and environments [4, 28, 29]. Research directions include video understanding and summarisation [28, 30, 31], hand-object interaction [32, 33], interpersonal interactions [34, 35], among others. Additionally, eyewear-based sensors also introduce novel research opportunities on eye-gaze understanding and prediction [36, 37, 38]. Smart glasses with camera and audio have recently seen commercial success [39], along with continued interest in AR/VR technology [40, 41].

**Egocentric Datasets.** Several open-source datasets and benchmarks have recently been introduced, including EPIC-KITCHENS [13] and Ego4D [12]. These datasets have highlighted the need for research on audiovisual diarization, episodic memory, action anticipation, localisation and tracking, and other domains. A large fraction of the published datasets focus on everyday activities, and indoor and outdoor tasks [13, 12, 42]. Recent datasets such as CharadesEgo[43], EgoExo4D [14], and EgoExoLearn [44] also utilise exocentric/allocentric perspectives, with a goal of improving understanding, enabling cross-view representation learning and skill transfer. Some industry-specific tasks, such as assembly-disassembly, goal step understanding, and skill assessment, have also been proposed [18, 14]. Benchmarking tasks, such as mistake detection, have also seen increased attention [18, 15, 16].

**Gaps and Opportunities.** In this paper, we study and present industrial contexts and potential cases in which an egocentric vision-based assistant can be expected to operate in the near future. Such an assistant would be required to understand the worker's surroundings and their personal context, guide the user through complicated environments and tasks, answer questions, detect mistakes, and reason about various factors with careful consideration [5, 4, 45, 46]. Datasets such as Meccano [17], Assembly101 [18], and HoloAssist [47] involve industry-like situations, with participants carrying out procedural tasks and following instructions. However, most of these tasks involve participants working at a workstation or a desk [18]. In real-world scenarios, the worker is needed to move from one place to the other, carry out long and arduous tasks, and collaborate with others. The recent introduction of the Project Aria Toolkit unlocks several opportunities for data collection and research [4, 48]. The Aria device is lightweight and mimics the form factor that future egocentric wearable devices are likely to have [4, 40]. This enables the users to move freely and capture context-rich multimodal data in their environment. Thus, participants can perform cognitively and physically demanding tasks, similar to a true industrial setup, while being able to move naturally and untethered from a working desk [49]. Secondly, the toolkit also allows users to work collaboratively on a given task, which can be studied in much greater detail via the multimodal sensor output. Figure 2 shows an example. Lastly, prior datasets focus on short to medium tasks [18, 47], leaving long-horizon activities (20+ min) underrepresented.

## 3 IndEgo Dataset

**Industrial Scenarios.** We categorise tasks into five broad groups: *assembly and disassembly* , *inspection and repair* , *logistics and organisation* , *woodworking* and *miscellaneous (others)* . There may be overlap, especially in longer tasks (e.g. the worker transports the device and tools to another location before disassembly); in such instances, we defer to the overall goal of the video for categorisation. The tasks are cognitively and physically demanding, requiring skill and effort in terms of reasoning and manipulation. Further details in Appendix A.

**Tools, Devices, and Machines.** The tools used for the work include common and domain-specific tools found in industry or research labs. For assembly/disassembly, we use mechanical devices,

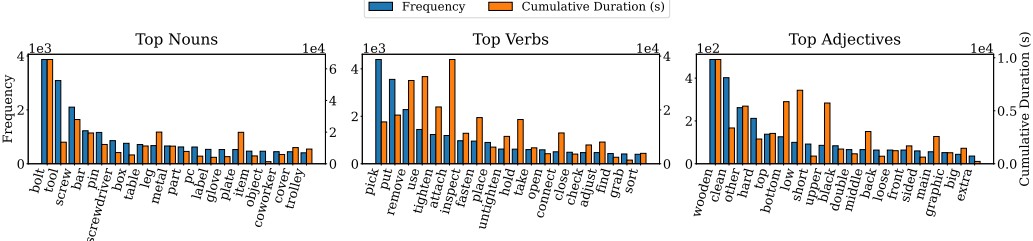

Figure 3: Grouped bar charts of frequencies (left axis) and durations (right axis) for the fine-grained action annotations: Top 20 nouns (left), verbs (middle), and adjectives (right). Our dataset covers diverse industrial contexts, which are not represented by current egocentric/exocentric datasets. This highlights the multimodality and human-centric attributes of IndEgo.

| Category | $T_{avg}$ | 1-Person | Collaborative | Narration | #Ego | $T_{Ego}$ (h) | #Exo | $T_{Exo}$ (h) |
|---|---|---|---|---|---|---|---|---|
| Assembly | 15.2 m | ✓ | ✓ | ✓ | 188 | 47.5 | 152 | 30.4 |
| Disassembly | 11.1 m | ✓ | ✓ | ✓ | 136 | 24.9 | 112 | 17.0 |
| Inspection and Repair | 7.8 m | ✓ | ✓ | ✓ | 238 | 30.9 | 202 | 17.7 |
| Logistics/Organisation | 4.5 m | ✓ | ✓ | ✓ | 456 | 35.4 | 158 | 8.1 |
| Woodworking | 7.5 m | ✓ | ✓ | ✓ | 148 | 18.4 | 116 | 14.9 |
| Miscellaneous | 1.5 m | ✓ | ✓ | ✗ | 378 | 9.4 | 352 | 8.7 |
| Tools/Objects in Context | 120 s | ✓ | ✗ | ✗ | 604 | 20.1 | – | – |
| Tools/Objects Demo | 53 s | ✓ | ✗ | ✓ | 302 | 4.5 | – | – |
| Singular Actions | 21 s | ✓ | ✓ | ✗ | 1010 | 5.9 | – | – |
| **Total** | 205 s | ✓ | ✓ | ✓ | **3460** | **197.1** | **1092** | **96.8** |

Table 1: A breakdown of the IndEgo dataset, showing the key categories and related statistics. $T_{avg}$ gives the average duration of the recording, #Ego gives the number of videos from the Egocentric perspective, $T_{Ego}$ gives the total cumulative time for egocentric data, #Exo gives the number of videos from the fixed exocentric perspective, $T_{Exo}$ gives the total cumulative time for exocentric data.

PC cabinets, and proprietary assemblies. Figure 9 shows some examples. A detailed description is available in Supplementary Materials.

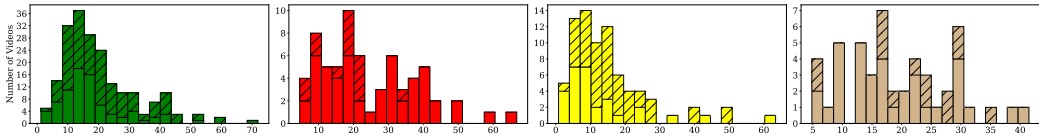

Figure 4: Histogram of egocentric video durations (minutes) of medium-longer task sequences for each category: assembly/disassembly , inspection/repair , logistics/organisation , and woodworking . The hatched regions represent two-person collaborative tasks. Details on the Miscellaneous category are provided in Appendix (A and K).

## 3.1 Data Collection Process

**Participants.** The data was collected by 20 participants, 15 male and 5 female. The selection was done in an unbiased manner, and the ratio reflects the natural skew seen in industrial work. They have varying degrees of experience in industrial skilled work (beginner to expert), and come from different nationalities and ethnic backgrounds. For our study, all participants used only English for demonstration and narration. Additional details in Appendix C.

**Hardware Setup.** Each participant used the Aria device [4] for the egocentric perspective. We defined a custom profile and recorded data from all available sensors for all our sessions. The camera sensors recorded data at 10FPS. The main RGB camera was set to the highest resolution (8MP: 2880

$\times$ 2880). For longer recordings, we use a portable powerbank, connected via a USB cable. With this setup, the maximum duration of a single recording was approx. 68 min (due to the on-device storage). Additionally, we use an external exocentric camera for several (1092) instances. We use different camera sensors (Sony A6400 APSC with Sigma 16 and 30 mm lens, Samsung Galaxy A51 and iPhone 16 as backup) with 1920 $\times$ 1080 resolution. The external camera is placed in the approximate position of a hypothetical third-person human observer, with a clear view of the participant and the working area. The raw data collected by the participants was stored and processed on a dedicated Workstation. Further details in Appendix E.

**Privacy and Ethics.** The participants were thoroughly informed about the general scope of the project and the intention to publish and open source the dataset. After obtaining written consent, they were given a primer on using the Aria device and the standard practices (work and safety, ethics, organisation of used tools, etc.). The working space is shared with other researchers, technicians and workers, who were all informed about the ongoing recording activities and protocol to avoid breach of their privacy. Written consent was also obtained from those who appeared in the background in the recordings. The project objectives, collected data, and other contents were examined by the reviewing authorities and approved for this submission.

**Protocol.** The activities took place at an industrial research facility over several months. Each session starts with a review of the planned tasks, followed by gathering the necessary tools/devices. Each participant calibrated the Aria device to accurately track their eye gaze (via the eye tracking cameras). The different tasks and scenarios were carried out in different settings: First, the participants were given a goal (e.g. *disassemble the mechanical frame*) without any specific steps or instructions. They were asked to carry out the task by narrating their inner monologue, including instances when they were not sure about a certain action or when they had made a mistake. Second, they received thorough, step-by-step instructions, as well as guidelines for confirming whether they had executed the task correctly. Thirdly, they were accompanied by a fellow participant for challenging activities where they collaborated on the task as a team. In the latter, both participants wore the Aria device and were asked to talk amongst each other and narrate their thoughts. We also organised scenarios where the participants had two identical setups and tools, and the goal was for the more experienced person to guide and teach the other person through the process via demonstration and dialogue. The participants had access to an attendant, who intervened only in cases of emergency or when the participants requested guidance. Lastly, the participants repeated certain sequences from the dataset with *mistakes* in various scenarios for the *Mistake Detection* (MD) benchmark (elaborated in Section 4).

They also collected data on *singular actions* under different circumstances. These are general and shorter actions with one specific goal (e.g. unloading a trolley, fastening a bolt, clamping a wooden block) which are likely to occur in different contexts in the industry. To further add variation, participants employ different objects (heavy/light, single/many), locations, and backgrounds. An additional set of recordings was done with the *tools and objects in context*, with the users interacting with/using the tools and objects. The focus is on the particular tool (w.rt. eye gaze, hands, etc.) in various circumstances. Finally, the participants recorded explanations and demonstrations (with narration) on how to use the different tools and their purpose. Table 1 gives a breakdown of the various segments of the dataset.

## 3.2 Annotation, Labelling, and Data Modalities

The participants annotate the data for their own recordings, which is then reviewed by a second person. For collaborative work, each user annotates their individual actions. Details on the justification and the established protocol are available in the Appendix (D and F). Figure 2 shows an example, where the ongoing procedure, and the actions of the two workers, is put into context. The actions are annotated into logical segments, e.g. *connect metal bar, attach drill bit to the drilling machine,* etc. We also annotate the keysteps for the procedural activities.

We share the raw recorded data from all sensors, along with the processed video files and the extracted frames. The processed outputs from the Aria recordings include eye gaze estimation, hand pose estimation, semi-dense point cloud and trajectory data. For procedural activities, the task graphs, instruction guides (originally made available to the participants) and metadata are available. Additionally, we provide extracted audio transcripts of the narrations and dialogue. Additional details in Appendix F.

| Dataset | Scenario | Hours (Ego) | Exo | Collaboration | Gaze | Motion | Narration | Actions | Keysteps | Mistakes | QA |
|---|---|---|---|---|---|---|---|---|---|---|---|
| EPIC-KITCHENS [13] | Kitchen | 100 | ✗ | ✗ | ✓ | ✗ | ✓ | ✓ | ✗ | ✗ | ✗ |
| CharadesEgo [43] | Daily | 34 | ✓ | ✗ | ✗ | ✗ | ✓ | ✗ | ✗ | ✗ | ✗ |
| Ego4D [12] | Multiple | 3670 | ✓ | ✗ | ✗ | ✓ | ✓ | ✗ | ✓ | ✗ | ✗ |
| LEMMA [50] | Daily | 10 | ✓ | ✓ | ✗ | ✗ | ✗ | ✓ | ✗ | ✗ | ✗ |
| Ego-Exo4D [14] | Multiple | 221 | ✓ | ✓ | ✗ | ✓ | ✓ | ✓ | ✓ | ✗ | ✗ |
| EgoExoLearn [44] | Daily, Lab | 120 | ✓ | ✗ | ✓ | ✗ | ✓ | ✓ | ✓ | ✗ | ✗ |
| Nymeria [51] | Daily | 300 | ✓ | ✓ | ✓ | ✓ | ✓ | ✓ | ✗ | ✗ | ✗ |
| AssistQ [47] | Assistive | 3 | ✗ | ✗ | ✗ | ✓ | ✓ | ✓ | ✗ | ✗ | ✓ |
| Meccano [17] | Industry-like | 7 | ✗ | ✗ | ✓ | ✗ | ✓ | ✓ | ✗ | ✗ | ✗ |
| HoloAssist [47] | Assistive | 166 | ✗ | ✓ | ✓ | ✓ | ✓ | ✓ | ✓ | ✓ | ✗ |
| Assembly101 [18] | Industry-like | 42 | ✓ | ✗ | ✗ | ✗ | ✗ | ✓ | ✓ | ✓ | ✗ |
| IndEgo (ours) | Industrial | 197 | ✓ | ✓ | ✓ | ✓ | ✓ | ✓ | ✓ | ✓ | ✓ |

Table 2: Comparison with related datasets on scenarios, modalities, and annotations. Hours (Ego) refers to the cumulative duration of distinct egocentric video recordings. **Top:** Datasets with diverse and everyday scenarios. **Bottom:** Datasets in an industry-like or assistive setting.

**Inter Annotator Agreement.** We conducted a brief study on a portion of the dataset (cumulative 3 hours of data, all scenarios). The recordings were annotated by up to 3 participants separately. For each annotation, we assess the agreement between the temporal annotations in each frame.

*Keystep Annotations:* We see an excellent agreement between the annotators, with a Krippendorff's Alpha of $\alpha = 0.97$. For these, the only divergence in agreement tended to be the start and end of a defined step (e.g. 210s vs 212s in a recording of 557s in total).

*Finegrained Annotations:* For a strict agreement (exact lemmatised verb + noun match for a given frame), we report a Krippendorff's Alpha of $\alpha = 0.25$. However, upon grouping similar verbs and nouns together (e.g. [detach, remove], [allen wrench, hex key], [get, grab, pick up], [colleague, coworker]), the value rises to $\alpha = 0.54$. We acknowledge there are certain variances between the way in which participants annotate and break up their actions, e.g. one participant annotated a temporal segment as *pick up device*, followed by *rotate device*, while another participant annotated the entire segment as *flip device.* We also see variance in the way the annotators describe an action, e.g. *search the drawer* and *find tool* were annotated for the same action. Similarly, *hold metal bar* and *assist coworker* were also annotated for the same action in a collaborative task. We believe that such differences are not a weakness, but rather reflect different ways in which an action may be interpreted.

### 3.3 Dataset Distribution and Analysis

The dataset comprises 197.1 hours of egocentric data, equating to 7.1M frames from the main RGB camera of the Aria device, alongside 13.9M SLAM frames and 13.9M eye-tracking frames, synchronised with the main RGB recordings. Additionally, the dataset includes 96.8 hours of exocentric recordings, corresponding to 10.5M RGB frames. Table 1 provides a detailed breakdown of the categories, including Mistake Detection (MD) tasks, which are generally of shorter duration, as well as medium-to-long industrial tasks. A further breakdown of these longer sequences is illustrated in Figure 4, which also highlights the distribution of collaborative recordings across four task categories. The miscellaneous category, primarily consisting of MD recordings, includes tasks that do not clearly fit within the other predefined categories.

Participants provided annotations for approximately 34k fine-grained actions in total, each containing task-specific verbs, nouns, and adjectives. Figure 3 visualises the Part-of-Speech (POS) breakdown across medium-to-long task sequences. The frequency of the top occurring POS and their cumulative duration across all categories is also shown. The median annotation duration is 5.7 seconds, with a minimum duration of 0.19s (e.g. *hand object over to coworker*) and a maximum of 214s (e.g. *observe demonstration from coworker*). Users also labelled key procedural steps for each sequence, providing structured keystep annotations to support task understanding and benchmarking.

### 3.4 Comparison with Current Datasets

Table 2 gives an overview of the attributes of IndEgo in relation to other egocentric datasets. The bottom portion shows datasets with a primary focus on assistive technologies and industry-like scenarios. Our dataset aims to fill a crucial gap w.r.t. industrial cases, collaborative work, as well as physically and cognitively demanding tasks. Moreover, its multimodality (video, audio, narration, gaze) adds to the research and application potential.

**Comparison with Project Aria-related datasets.** Multiple datasets collected using the Project Aria device and toolkit have been released, including Ego-Exo4D [14], Aria Everyday Activities [42], Aria Digital Twins [48], Aria Synthetic Environments [52], HOT3D [32] and Nymeria [51]. All these datasets harness the multimodal sensor capabilities and the data processing tools available. Our dataset is unique in terms of its focus on industrial cases and assistance scenarios. Further details in Appendix H.

# 4    Benchmarks and Evaluation

A dedicated workstation (Nvidia A6000 GPU) is used for all experiments. We use consistent hyperparameters for all models (details in SM). Abbreviations for the models used; VL3: VideoLLaMA3 8B [53], IVL2.5: Intern-VL-2.5 (InternViT-300M-448px-V2_5) [54], QVL2.5: Qwen2.5-VL-7B [55], GFT: Gemini 2.0 Flash Thinking Experimental [56]. * via API, Experimental, model architecture unspecified. All models use 480p video@10FPS, further details in Appendix I.

## 4.1    Mistake Detection.

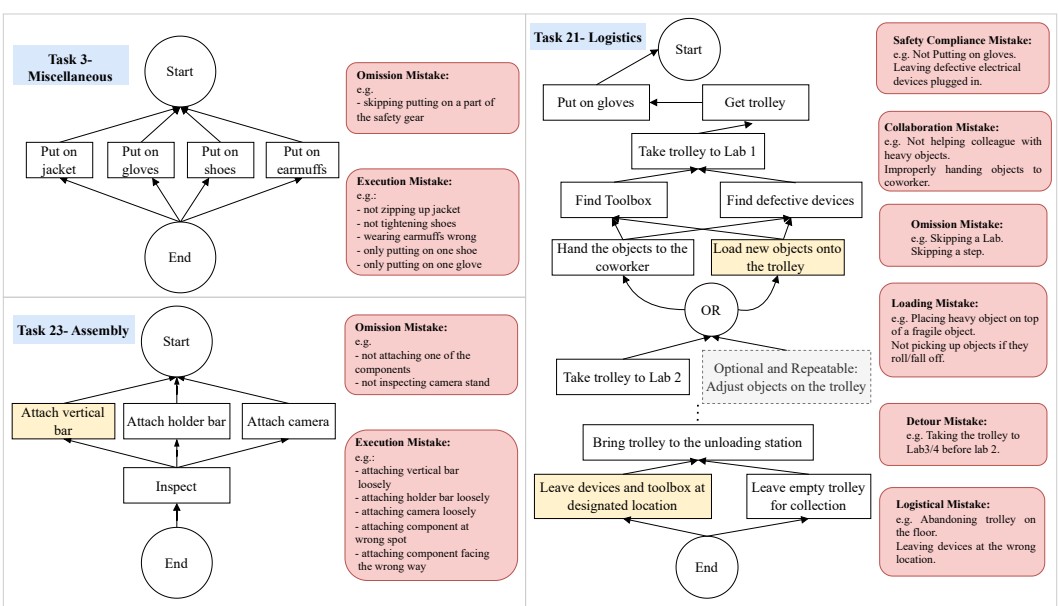

Figure 5: Task graphs for some scenarios from the MD benchmark, along with some commonly seen mistakes (not exhaustive). The flow of actions is from top to bottom, and dependencies are shown with an arrow. ☐ denotes labour-intensive steps. **Left:** *Miscellaneous* sequence **#3** at the top, and *Assembly* sequence **#23** at bottom. **Right:** *Logistics* sequence **#21**. The graph is shortened due to spatial constraints.

We recorded 1166 egocentric sequences across 25 tasks and all five categories, with correct actions and *planned + unplanned mistakes*. These involve procedural tasks (e.g. *repairing a PC*) as well as non-procedural tasks (e.g. *tidying the workspace after finishing work*). The tasks include at least 42 recordings each, with an average of 18 correct and 24 with *mistakes*. These involve common types of errors, such as *skipping* a procedural step, doing things in the *incorrect order*, and *adding* unnecessary steps. We also add *industry-specific mistakes*, such as not following workplace safety guidelines, mishandling tools/devices, and misplacing items. Additionally, for collaborative tasks, we include mistakes such as carelessly handing objects to a coworker and not helping a coworker carry out a physically strenuous task. Figure 5 shows an example of a collaborative task, with some examples of mistakes. We provide exocentric recordings for most tasks. We find an that exocentric perspective can provide additional context for the steps in case of limited egocentric visibility (e.g. *wearing PPE/safety kit*). For annotations, we provide the step-wise action segments and mistake labels, along with the descriptions of the mistakes. We also provide task graphs for the correct sequences.

| | Approach | P | R | F1 | F1$^S$ | F1$^{PF}$ | F1$^{IF}$ | F1$^H$ |
|---|---|---|---|---|---|---|---|---|
| ZS | VL3 [53] | 15.6 | 46.2 | 23.3 | 36.2 | 38.2 | 27.4 | 32.1 |
| ZS | IVL2.5 [54] | 16.2 | 48.2 | 24.2 | 38.1 | 37.1 | 29.0 | 33.2 |
| ZS | QVL2.5 [55] | 15.9 | 50.1 | 24.1 | 38.8 | 36.5 | 28.8 | 34.1 |
| ZS | GFT* [56] | 35.6 | 48.2 | **40.9** | 51.2 | 42.2 | 34.7 | 48.0 |
| MLP | VL3 [53] | 30.4 | 56.7 | **39.5** | 48.1 | 38.8 | 32.1 | 41.3 |
| MLP | IVL2.5 [54] | 31.6 | 50.0 | 38.7 | 47.7 | 39.1 | 30.5 | 42.2 |
| MLP | QVL2.5 [55] | 31.4 | 51.6 | 39.1 | 42.6 | 39.8 | 35.4 | 44.0 |
| Tr | VL3 [53] | 34.5 | 33.3 | 33.9 | 39.2 | 35.5 | 29.1 | 38.5 |
| Tr | IVL2.5 [54] | 30.1 | 41.7 | 35.5 | 36.5 | 38.7 | 32.1 | 39.2 |
| Tr | QVL2.5 [55] | 33.3 | 41.0 | **36.7** | 37.0 | 39.4 | 29.5 | 36.7 |
| MLP | VL3 [53] (EM) | 21.3 | 55.0 | 30.7 | 36.2 | 38.2 | 30.1 | 32.2 |
| MLP | IVL2.5 [54] (EM) | 23.3 | 49.2 | 31.6 | 35.2 | 32.7 | 31.6 | 30.5 |
| MLP | QVL2.5 [55] (EM) | 24.1 | 51.0 | **32.7** | 34.2 | 32.0 | 32.1 | 40.1 |

Table 3: Baseline Results for Mistake Detection. **Top:** Zero-shot evaluation - the F1$^S$ and F1$^H$ scores are higher because the model was prompted for the particular mistake. **Middle:** Fine-tuning binary classification (Mistake/Correct) via an MLP. and Transformer. **Bottom:** Early Mistake Detection (EM), where only 50% of the initial frames of the segment are available for the prediction.

In industrial settings, certain mistakes can be more consequential/critical than others. Hence, we add context-dependent categorisation for such errors. These include Severe Mistakes (S) - which can lead to unnecessary disruptions (e.g. mishandling a fragile object); mistakes leading to failure of the entire process (PF) - e.g. placing the wrong item in a shipment; mistakes that impact all future steps (IF) - e.g. forgetting to open the trolley hatch before loading material; and mistakes posing a risk of physical harm to self or other (H) - e.g. cleaning a wound with a dirty cloth. The majority of mistakes in the dataset do not fall into either category; 2.3% are S, 18.7% are PF, 7% are IF, and 5% are H. Further details are provided in Appendix K.

| | Approach | P | R | F1 | F1$^S$ | F1$^{PF}$ | F1$^{IF}$ | F1$^H$ |
|---|---|---|---|---|---|---|---|---|
| Ego | VL3 [53] | 17.1 | 48.0 | 25.2 | 34.1 | 37.2 | 28.4 | 35.5 |
| Ego | IVL2.5 [54] | 18.2 | 48.7 | 26.5 | 32.3 | 36.1 | 30.1 | 34.2 |
| Ego | QVL2.5 [55] | 16.5 | 50.5 | 24.8 | 34.1 | 29.1 | 30.5 | 32.0 |
| Ego | GFT* [56] | 36.5 | 47.2 | **41.1** | 50.1 | 43.2 | 33.6 | 44.5 |
| Exo | VL3 [53] | 20.1 | 44.2 | 27.6 | 34.7 | 34.8 | 31.2 | 29.1 |
| Exo | IVL2.5 [54] | 18.7 | 48.8 | 27.0 | 37.5 | 33.3 | 29.8 | 32.5 |
| Exo | QVL2.5 [55] | 21.1 | 49.6 | 29.6 | 32.4 | 29.4 | 31.4 | 32.6 |
| Exo | GFT* [56] | 35.1 | 51.1 | **41.6** | 48.5 | 41.0 | 34.3 | 46.6 |

Table 4: Zero-shot evaluation for MD on the first 15 tasks. **Top:** Egocentric Perspective. **Bottom:** Exocentric Perspective. A similar trend is seen across the two views.

Table 3 gives baseline results on the MD benchmark. For zero-shot evaluation, the VLMs are presented with the video and asked to predict whether it contains the correct action or a mistake. For the test, users designed specific prompts that instructed the VLM to look for specific factors (e.g. *did the user use the appropriate tool for the task?*), and also to check if a known mistake is occurring (e.g. *did the user leave the trolley in the corridor?*). The cluttered background and industry-specific nature of the data make it difficult for the VLMs to accurately reason about the action.

For finetuning results, we extract the feature embeddings for the action steps from the VLMs, which are then forwarded to either an MLP classifier or a Transformer (Tr) based classifier. Steps with *mistakes* are assigned a higher weight (5). For early mistake detection (EM), only the initial 50% of frames are available to the model during validation/testing. Table 4 compares the zero-shot evaluation results on the first 15 MD tasks for the egocentric and exocentric perspectives.

| Model | Ego | Exo | Ego + Exo |
|---|---|---|---|
| GFT (ZS) [56] | 0.43 | 0.39 | 0.44 |
| VL3 + Tr [53] | 0.33 | 0.32 | 0.37 |
| IVL2.5 + Tr [54] | 0.29 | 0.30 | 0.33 |

Table 5: Mistake detection F1 scores for Ego-only, Exo-only and joint-views. We compare zero-shot (ZS) and finetuned (+Tr denotes a transformer block) models. The joint view consistently yields the best performance.

We evaluated mistake detection on a subset of 10 tasks to assess the value of joint ego- and exocentric views, reporting F1 scores in Table 5. Our study included zero-shot (ZS) evaluation and a finetuning approach where we extracted step-wise embeddings from a VLA, concatenated the corresponding view embeddings, and trained a transformer-based model (+Tr) with k-fold cross-validation. The joint `Ego + Exo` view slightly outperforms single-view inputs across all models. A qualitative review of the zero-shot results indicates the exo view is most beneficial when the ego view is occluded or the mistake is out of focus, though in some cases it offered no improvement or led to incorrect predictions.

| Model | RGB only | RGB + Audio | RGB + Gaze | RGB + Audio + Gaze |
|---|---|---|---|---|
| GFT (ZS) [56] | 0.38 | 0.41 | 0.39 | 0.42 |
| VL3 [53] | 0.27 | 0.26 | 0.28 | 0.30 |
| IVL2.5 [54] | 0.30 | 0.28 | 0.29 | 0.29 |

Table 6: F1 scores for *Mistake Detection* (averaged across 10 tasks), ablating audio and gaze modalities. Our investigation into the results for each *task* shows that the utility of each modality is highly context-dependent.

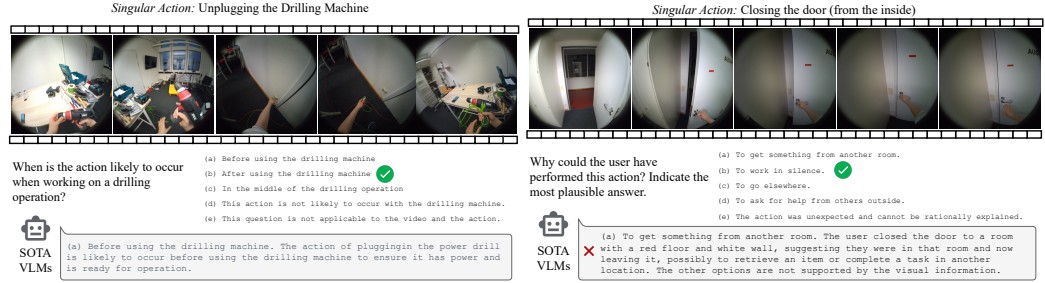

Figure 6: Example questions from the reasoning-based QA benchmark from the dataset, followed by an answer produced by a SOTA model [53]. **Left:** The Answer was wrong, potentially because the model misunderstood the temporal action (attaching vs. detaching). **Right:** The answer is wrong because of incorrect situated reasoning (closing the door from the inside is different from walking out and closing the door).

**Ablation Study on Modality Contribution.** We conducted an experiment on the mistake detection portion of the dataset (10 tasks across all scenarios), reporting average F1 scores in Table 6. For zero-shot evaluations, eye gaze was overlaid on the RGB stream, and the model was prompted to focus on the indicated region. Our qualitative review reveals that a modality's impact is context-dependent. For instance, removing the audio modality in noisy environments can slightly improve performance. Conversely, in collaborative tasks, audio provides a crucial signal for understanding context. Similarly, prompting the model to use eye gaze is beneficial when a user performs an action incorrectly within their field of view. However, when the mistake occurs outside the gaze region (e.g., forgetting to pack an object in a container), adding the gaze modality can slightly worsen performance.

## 4.2 Reasoning-Based Video Question Answering

It is essential for egocentric assistants and agents to be able to understand the scene, the task, and reason about the implications of various actions. Current state-of-the-art (SOTA) multimodal models have demonstrated remarkable ability w.r.t. processing text and image data [57, 58, 21, 22]. There has also been an increased focus on reasoning and planning [45, 59, 60]. Recent Video Language Models (VLMs) have been shown to understand allocentric videos, summarise and answer questions [61]. These capabilities also generalise to egocentric and industrial scenarios. We curated a set of 3105 question and answer pairs, based on the *singular actions, fine-grained annotations* (2020) and the other categories (1085). The questions require visual perception and reasoning about the user's action and the objects in the field of view. The questions are designed so that an average technician (or a layperson with access to the internet) can answer them. Figure 6 shows an example of two questions based on the *singular actions*. The questions can be broadly split into four groups: those focusing on the temporal aspect of the action (Tm - 14%), those requiring situated reasoning (Si - 28%), those requiring visual recognition of the objects in the FOV (Re - 32%), and those requiring analogical or abductive reasoning (A - 26%).

Table 7 gives the result (% accuracy) for SOTA VLMs on the *singular actions* portion of the benchmark. As seen in Figure 6, questions that can be readily answered by humans, could be challenging for VLMs. For comparison, the questions were also answered by human evaluators (4).

| Model | Acc$^{Tm}$ | Acc$^{Si}$ | Acc$^{Re}$ | Acc$^{A}$ | Acc |
|---|---|---|---|---|---|
| VL3 [53] | 52.2 | 60.3 | 59.4 | 57.5 | 58.2 |
| IVL2.5 [54] | 51.7 | 61.1 | 58.2 | 56.0 | 57.6 |
| QVL2.5 [55] | 53.2 | 60.8 | 59.3 | 56.5 | 58.1 |
| GFT* [56] | 55.4 | 62.1 | 67.2 | 68.3 | 64.1 |
| ML2* [62] + Label | 92.3 | 51.4 | 42.8 | 78.3 | 61.4 |
| Human | 92.6 | 89.6 | 90.4 | 88.6 | 90.0 |

Table 7: Zero-shot evaluation of the reasoning based *QA benchmark* on singular actions portion of the dataset with some SOTA VLMs. Mistral-Large2 model was given the action label (e.g. closing the door from the inside) and asked to select a correct answer. Human evaluators had access to the industrial tool database from the dataset. *via API, Experimental, model architecture unspecified.

### 4.3 Task Understanding in a Collaborative Setting

It is important for egocentric assistants to understand the actions of the wearer, as well as others, esp. when working together on a common task. We provide extracted action pairs from the dataset for procedural and non-procedural tasks. The goal is to differentiate and predict the actions of the user and the coworker for the given segment, and understand their relative role (e.g. collaborator, teacher/student). A zero-shot evaluation yields an accuracy of 35.2% (GFT), with the VLM struggling to differentiate between the egocentric viewer's actions and the actions of the coworker. Employing a similar setup as MD (VLM embeddings forwarded to a classifier). We train the model to predict what the wearer would do based on the actions of the coworker, and report a baseline accuracy of 42.1% (VL3 + Tr). Both evaluations were performed on approximately 50% of the collaborative egocentric recordings from each scenario. Ablation studies show, that removing the audio modality negatively impacts the zero-shot results. Additional details are provided in Appendix (B and M).

### 4.4 Additional Experiments

**Summarisation, Long Video Understanding and Reasoning-QA.** The goal is to produce a coherent summary of the input video (text and image), with minimal loss in context and key events. As ground truth, we sample the egocentric recordings based on the keysteps and fine-grained annotations. The dataset contains up to 68 min long videos. The longer videos tend to belong to the *unguided* sequences (e.g. the user is given a repair task without any instructions). These scenarios reflect the real-world cases where an egocentric agent could be beneficial. The baseline evaluation approach is presented in the Appendix. The reasoning-based QA task for the longer tasks (1085 questions) is designed to assess this aspect. Audio-only (transcript from user narration) yields a baseline accuracy of 67.3%. Additionally, we provide summaries and context on the tasks performed by the participants, along with additional comments (e.g. unplanned mistakes, incorrect practices). For collaborative tasks, we provide summaries from both points of view and an overall summary.

**Additional Modalities.** Certain tasks and actions are inherently multimodal (e.g. attaching a battery to the drill produces a *click* sound). Additionally, other modalities (user trajectory, gaze) and the exocentric perspective offer a rich array of research possibilities, including cross-view and cross-modal alignment.

## 5 Conclusion

We introduced *IndEgo*, a unique multimodal dataset for egocentric and exocentric vision in industrial tasks, including assembly/disassembly, logistics organisation, inspection, repair, and woodworking. IndEgo consists of 3,460 egocentric recordings (approximately 197 hours) and 1,092 exocentric recordings (approximately 97 hours), along with eye gaze, narration, audio, motion, and other sensor data. We provide detailed annotations and challenging benchmarks for procedural and non-procedural task understanding, Mistake Detection, and reasoning-based Question Answering. Baseline results on Mistake Detection, Question Answering and collaborative task understanding show that the dataset presents a challenge for the SOTA multimodal models, highlighting the need for further exploration. IndEgo consists of diverse cognitively and physically intensive industrial tasks, filling a relevant gap in egocentric vision research. We believe it will be a valuable resource for advancing AI-assisted industrial applications and enhancing productivity, safety, and efficiency in real-world operations.

**Acknowledgments**

This work is supported by the German Federal Ministry of Research, Technology and Space (BMFTR) and the German Aerospace Center (DLR) under the KIKERP project (Grant No. 16IS23055C) within the KI4KMU program. We are grateful to the Meta AI and Reality Labs teams for the Project Aria initiative, including the research kit, associated tools, and services. We also thank Hugging Face for providing a public-dataset storage grant that enables large-scale hosting and community access to the IndEgo dataset. Data collection was conducted at the research labs and test field of the Institute of Machine Tools and Factory Management (IWF), TU Berlin. Finally, we extend our sincere thanks to all student volunteers and workers who contributed to the data collection.

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

# Appendix: Table of Contents

# A   IndEgo: Dataset Scenarios and Categories

**Motivation.** Industrial environments present a highly structured yet dynamic setting where egocentric AI can offer tangible value in assisting human workers, monitoring task execution, and enabling skill transfer. Unlike consumer or household egocentric datasets, our focus is on procedural tasks involving tools, assembly steps, and spatial reasoning. These are core elements of industrial workflows. The motivation for this dataset stems from the need to capture authentic, real-world sequences of skilled and unskilled labour, where timing, sequence adherence, and tool usage are critical. Our dataset is unique in its multimodal nature, combining egocentric and exocentric perspectives, speech, gaze, and action annotations across a diverse range of industrial tasks. It provides long-horizon activities, multi-user collaboration, and realistic variation in skill level and behaviour, all of which are underrepresented in existing datasets. By releasing this data and associated benchmarks, we aim to catalyse research in instruction following, mistake detection, human-AI collaboration, and embodied AI grounded in practical applications, offering a valuable foundation for future work in both academia and industry.

The industrial application scenarios were selected to represent a range of activities and settings that egocentric assistants and embodied agents are likely to encounter in the near future [46]. The tasks contain activities and actions, that are not covered by other egocentric datasets. Additionally, they require cognitive and physical effort on the part of the participants. Our aim is that these scenarios will help in general vision and Artificial Intelligence (AI) research and have a broader impact beyond industrial, and egocentric domains.

There were several challenges w.r.t. expanding our data collection to other industrial settings (including privacy, data protection and IP), which is why we decided against it. However, we performed data collection activities at several different locations on the facility, including noisy shop-floor like environments [19, 63]. Additionally, we intentionally collected data during different times of the day, with varying lighting, background activities etc. We acknowledge, that our work does not address all possible scenarios and challenges that could be seen in the industry [64]. However, we are optimistic that this will serve as a catalyst for increased attention and further exploration of this domain.

## A.1   Application Scenarios

We describe the five general categories below:

- **Assembly/Disassembly:** These involve the participants assembling and disassembling the devices, machines, and proprietary setups. Figure 9 shows some devices used for the tasks. These were chosen to be challenging tasks and get the participants to carefully plan their actions before execution. Most disassembly procedures are accompanied by a corresponding assembly recording. However, the two are treated as separate tasks and are stored, processed and annotated separately.

- **Logistics and Organisation:**  These involve the users collecting, carrying, transporting and organising tools, devices, and objects in industrial contexts. Figure 9 shows some devices used for the tasks. These tasks often involve users planning their activities and following safety guidelines before carrying out the steps. Organisation tasks include cases where the user has to search, arrange and put items away. Examples include user searching for tools before an activity, or returning them back after use.

- **Inspection and Repair:** This scenario involves the participants checking devices/equipment and repairing them, aiming to restore their proper function and form. We also use the items shown in Figure 9, along with some additional devices. The issues with most electronic equipment were preplanned and set by the assisting person, without prior knowledge of the participant responsible for inspecting and repairing it. W.r.t. mechanical devices, the flaws are either visual (e.g. the participants are given reference images of an ideal assembly/device, and are asked to correct defects) or functional (e.g. moving the crank does not equally move the legs of a mechanical frame).

- **Woodworking:**  This category is meant to include actions, objects, and activities that are not generally seen in other scenarios. The users perform basic operations, such as rasping,

drilling, attaching brackets, attaching two pieces together, and also more challenging cases, involving putting together a wooden box.

- **Miscellaneous:** This is a broad category, including all tasks and recordings that do not definitively fall into the other category. This includes general actions such as setting up a tripod and camera, packaging items in a box, as well as industry specific actions, such as wearing/removing PPE and administering first aid. Several actions in this category are of shorter duration, and belong to the *Mistake Detection* tasks, i.e. they were planned and designed to collect data on the *correct* and *erroneous* processes in several settings. Further details in Appendix K

**Categorisation.** There is natural overlap between the categories, e.g. tasks involving assembly/disassembly may require the participant to put on gloves and safety equipment, or putting together a wooden box could also be seen as an assembly task. In such situations, we categorise the recording based on the overall goal of the task and the key actions involved (e.g. repairing a PC also involves disassembly steps, but the goal is to get the device in a working state).

**Procedural vs. Non-Procedural Activity.** The five scenarios include procedural as well as non-procedural activities. We define procedural activity as a task involving separate steps with temporal dependencies, where doing things out of order can often lead to failure. Examples include assembly/disassembly. Non-procedural activities are tasks involving a set of actions that can all be independently carried out, with no rigid temporal dependencies and no specific order. Examples include wearing PPE (gloves, vests, safety shoes, etc.) and arranging a toolbox.

## A.2 Additional categories

The dataset also contains other categories of recordings (mentioned in Table 1). We describe them below.

- **Tools/Objects Demo:** Recognising and reasoning about the correct tool for a task is a fundamental challenge for AI assistants, particularly for small or domain-specific objects [19, 65]. To address this, our dataset involves users demonstrating and explaining how a particular tool/object is used. We compiled a list of 302 frequently used tools and devices, that were used in the application scenarios. The participants then recorded separate demo videos using only the Aria device (no exocentric view). These involve user narration and all other modalities with the Aria recording profile. The set contains videos with approx. 4.5 hours of cumulative time. Figure 9 shows examples of the tools and objects.

- **Tools/Objects in context:** This involves the same 302 objects in their everyday use/operation. The recordings are approx. 2 min long each, recorded by 2 separate participants in separate settings. The participants maintain their focus (eye gaze) on the object in question. There is no audio narration and no exocentric view. The recordings are intended as a supplement to the rest of the dataset, with an aim of training and testing ML models. These involve 604 recordings with approx. 20.1 hours of cumulative time.

- **Singular Actions:** These are short recordings that involve the participant carrying out a specific predefined task, e.g. putting on gloves, attaching a drill bit to a drilling machine, etc. The tools used belong to the 302 objects, however, the focus is on the given action, its meaning, and situated understanding based on the objects and the surroundings. We present the **Reasoning-based Question Answering (QA)** task on these action videos. The actions last between a few seconds to just under a minute. We include a total of 1010 recordings with approx. 5.9 hours of cumulative time.

## B  Collaborative Work

We broadly describe collaborative work as an activity that employs multiple agents who proactively carry out actions in a given environment to successfully accomplish a predefined goal. It is common in industrial scenarios to work on challenging tasks together, esp. for physically strenuous activities. The IndEgo dataset involves maximum 2 people working together at a given time (both wear the Aria device). We include the following types of collaborative work:

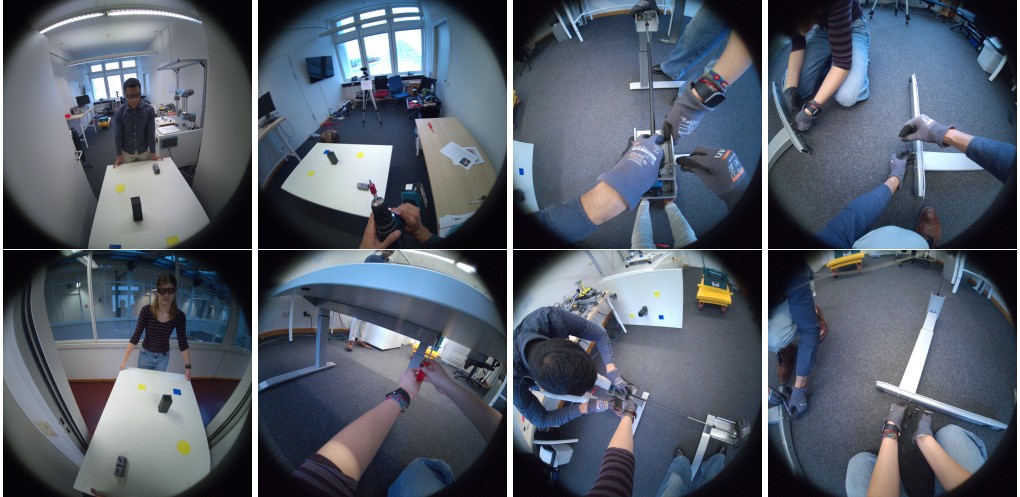

Figure 7: Example of a scenario in the IndEgo Dataset where two workers carry out a *disassembly task*. **Top:** Worker 1 perspective. **Bottom:** Worker 2 perspective for the corresponding frames in the sequence. Anonymised using EgoBlur [66].

- **Coworking:** This involves two users planning and carrying out activities across different application scenarios, where they work as equal partners to complete the task. The participants proactively assist and support each other in activities. This is the predominant type in our dataset. Figure 7 shows an example, which is also shown in Figure 2 in the main paper.

- **Supervision:** This involves an expert supervising and guiding the other participant on the task. The supervisor occasionally also helps the participant if they are stuck or need manual assistance (e.g. holding the attachment for assembly).

- **Teacher-Student:** This has similar roles as the previous type, however, the two people each have their own identical setups. The teacher demonstrates the task step by step, and the student observes, follows and asks questions as needed. Figure 8 shows a woodworking example.

## C Data Collection Contributors

The data collection process was carried out by 20 participants over a period of several months. As mentioned in the paper, the authors explained the framework and the goals of the project to the participants and obtained written agreement for collecting, storing and publishing the dataset. Table 8 gives the key attributes of the participants. The recordings were anonymised, i.e. researchers or engineers outside the data collection team are not aware of the details of the people involved in a given recording.

The participants come from different backgrounds; this can be observed in their narration, approach to tasks, etc. We aimed at addressing different scenarios and conditions (e.g. shop floor environment, working room environment, different times of the day, lighting, background noise). While the number of participants is modest, participation was entirely voluntary, and the primary focus of this work is on capturing diverse industrial tasks and procedures rather than demographic coverage.

The authors express their sincere gratitude to all individuals and groups who contributed to the data collection and annotation process for the *IndEgo* dataset. The following contributors (listed alphabetically to ensure fairness) have given their consent to be publicly acknowledged for their efforts. We also wish to thank other participants who contributed to the data collection, but have chosen to remain anonymous.

- Aftab Ahmad Arif (Technical University of Berlin).
- Kian Khalifehgholi (Technical University of Berlin).

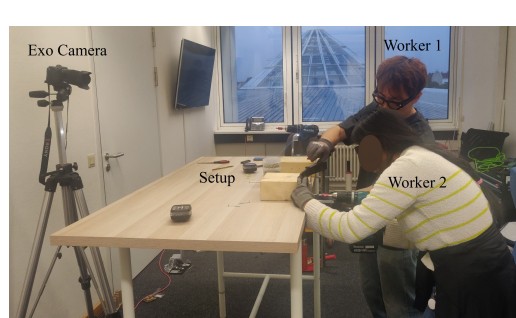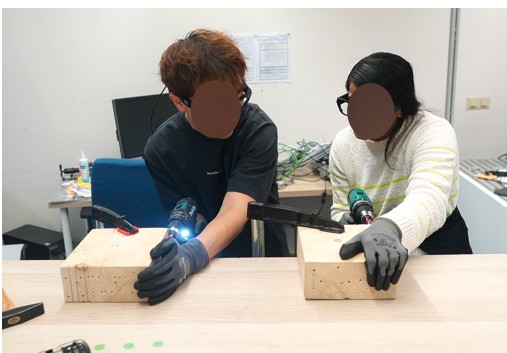
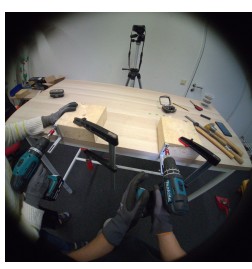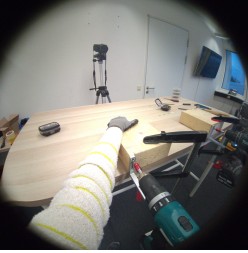

Figure 8: **Top:** *(left)* Example of a scenario in the IndEgo Dataset where one worker (*teacher*) demonstrates and teaches the other worker how a task is carried out. The *student* observes and follows the steps on their own identical setup, asking questions and seeking clarifications, if necessary. *(right)* View from the exocentric camera. **Bottom:** Egocentric perspectives of the *teacher* (left) and the *student* (right). Anonymised using EgoBlur [66].

| Parameter | Value |
|---|---|
| Gender (self-reported) | 15 male, 5 female |
| Nationalities | 10 (Europe, Asia, Middle East) |
| Age | Min. 19 Years \| Max. 46 Years
Average: 27.2 Years
Median: 24.0 Years |
| Work Experience | 0 to 24 years |
| Expertise (self-reported) | Novice (9)
Semi-proficient (7)
Proficient (4) |

Table 8: General details about the participants involved in the data collection and annotation process. The personal details of the individuals are anonymised.

- Lina Rost (Unaffiliated).
- Nazmi Kayan (Technical University of Berlin).
- Pengtao Xie (Technical University of Berlin).
- Qianshun Zhu (Technical University of Berlin).
- Xingyu Shang (Technical University of Berlin).
- Yanqing Luo (Technical University of Berlin).
- Employees of the *Werkstatt* at Fraunhofer IPK.
- Technicians at Institute of Machine Tools and Factory Management *(IWF)*.

In addition to the contributors listed above, several of the paper's co-authors were also significantly involved in the data collection and annotation efforts.

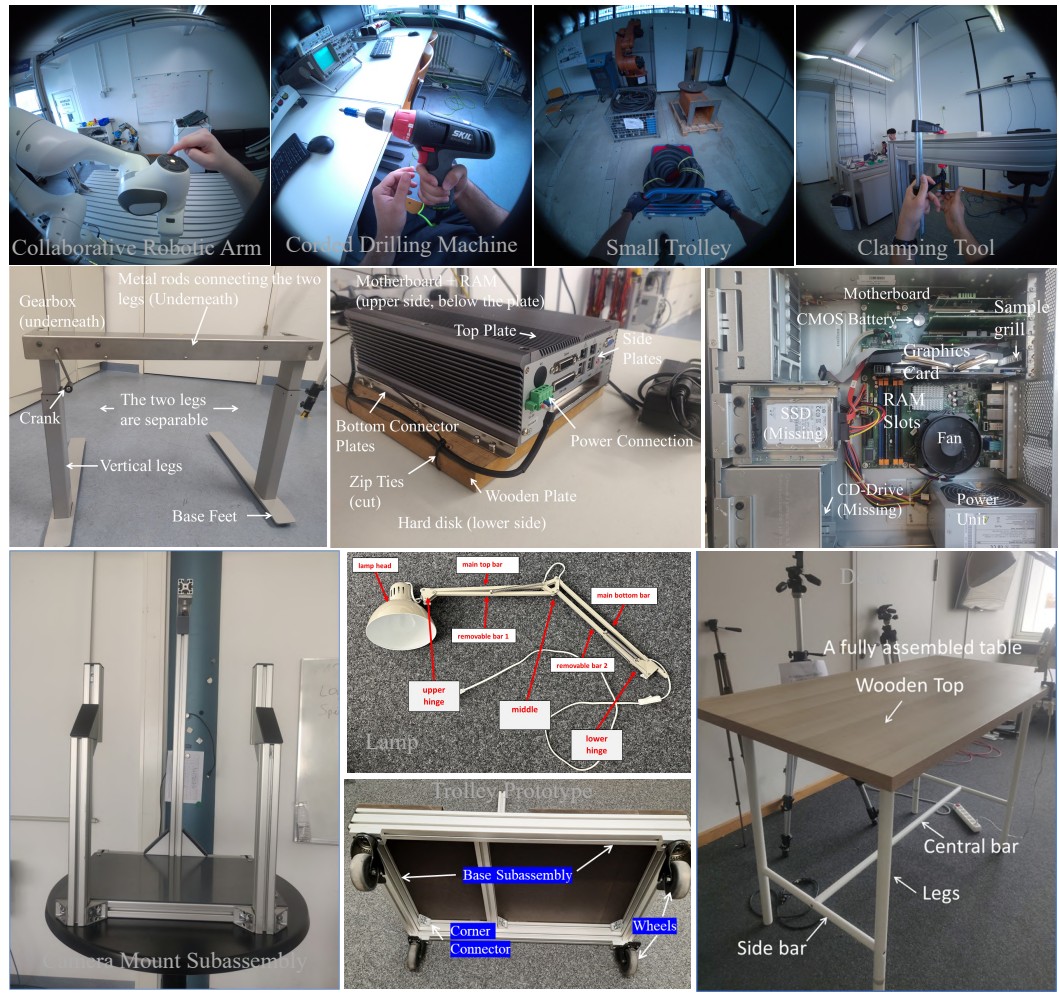

Figure 9: **Top:** Some commonly used tools and objects in various scenarios in the dataset. **Bottom:** Devices and Assemblies used for *assembly/disassembly* and *repair* tasks. The images are taken from the instruction document provided to the participants for the *guided* tasks. Additional details are available in the Supplementary Material

# D Data Collection Protocol

The participants were provided with a working space, tools and the target device for the various tasks. They started out with eye gaze calibration and set-up of the Aria device, while others set up the exo-cameras. The participants collecting the data start their individual recordings in sync with the exocentric recording/s. After the recording starts, the participants are free to plan and carry out the activities at their own pace (e.g. one PC repair task took 9 min to 58 min, for the same base setup). Additionally, they are free to move around the facility if they want to fetch another tool or find an object. They are advised not to mention their personal information or data while recording, and also not to discuss things unrelated to the task at hand. They are accompanied by an assistant, who can support them with troubleshooting issues, if any. We categorise the tasks as follows:

- **Unguided Tasks:** The participants have a general understanding of the task and the intended goal (e.g. repair the device, assemble the setup). However, they are not given any specific instructions. They are supposed to think step by step and come up with a plan for their work. They are allowed to correct any self-identified mistakes on the fly.

- **Guided Tasks:** The participants have access to an instruction document, explaining the correct steps and visual examples of the correct setup. They are free to change their actions and diverge from the document, based on their reasoning.

Furthermore, we include 3 scenarios w.r.t. the user narration:

- **Narration (Train of Thought):** The participant/s explain their train of thought in detail, including all general and specific actions, e.g. *turning on the lights, finding another screwdriver*. We include this for solo as well as collaborative work.

- **Narration (Conversation):** This applies to collaborative work. The users are free to discuss their actions and plans freely, as they generally would when working on a task.

- **No Narration:** Mostly applies to solo work. The user does not explain their actions. However, audio as a modality is still present. We included this to assess reasoning and understanding in AI models, without overreliance on the narrated elaboration.

## E    Hardware Setup

In this section, we describe the devices, sensor specifications, and other hardware details related to the dataset and the paper.

### E.1    Project Aria Device

This is the central focus of the dataset. The Aria device [4] consists of several sensors, which can be configured and switched ON/OFF based on the requirements. We use a custom recording profile for all egocentric recordings, the details are described in Table 9. The Aria Toolkit enables the output from all sensors to be automatically synchronised and calibrated without user intervention. Multiple Aria devices were used for the data collection.

| Parameter | Value |
|---|---|
| RGB Camera Resolution | $2880 \times 2880$ (8 MP) |
| RGB Camera Frame rate | 10 FPS |
| SLAM Camera Resolution | $640 \times 480$ |
| SLAM Camera Frame rate | 10 FPS |
| Eye Tracking Camera Resolution | $320 \times 240$ |
| Eye Tracking Camera Frame rate | 10 FPS |
| Microphones | ON |
| IMUs | ON |
| Magnetometer | ON |
| Barometer | ON |
| GPS | ON |
| Wi-Fi | ON |
| Bluetooth | ON |

Table 9: Details of the custom recording profile used for the Aria Devices. For additional details on the sensors, please refer to the preprint about the device [4].

We decided to collect egocentric data with 10FPS (main RGB, eye gaze, SLAM sensors) and maximum allowed resolution ($2880 \times 2880$ for the main RGB camera). There had to be a trade-off between the data resolution, frame rate and the storage requirements. We agree that the frame rate does not allow sub-second interactions in some cases. However, from our observation, this did not limit the applications and analysis potential of the dataset. The focus of our dataset is on a diverse set of finegrained actions, which range from 0.2s to several seconds. The annotation tool (VIA) also permits finer control over the temporal annotations (e.g. an action can start at 10.15s).

Additionally, we conducted a short study, where we took 10 tasks from the Mistake Detection portion of the dataset and reduced the frame rate down to 5 FPS. This resulted in a slight drop in performance (3% drop in F1 score with Gemini 2.0 Flash thinking, zero shot evaluation), however, most tasks were interpreted in the same manner as the 10 FPS baseline.

## E.2  Exocentric Cameras

The decision to include a third-person exocentric/allocentric camera perspective was done to provide additional context on industrial scenarios, esp. for tasks involving collaborative work, physically demanding work and first-person actions, such as wearing Personal Protective Equipment (PPE). The IndEgo dataset includes 1170 total exocentric recordings, across all key scenarios. The camera was set up on a tripod, focusing on the user and the operating area, from an approximate vantage point of a stationary observer. Figure 8 shows an example. Some collaborative cases involve dual exocentric perspective, especially because a single view would periodically get occluded by a participant operating on the device. The *Logistics and Organisation* scenarios have the lowest share of exocentric recordings, since they mostly involve participants moving between different labs and workstations. Table 10 gives the details on the devices used for these recordings.

Given the dynamic nature of the tasks, it was challenging to set up exocentric cameras. We aimed for a balance between capturing sufficient detail of the device/object of interest, and capturing user movement and background details. For tasks where the user can focus on a dedicated workspace (e.g. repair of a device), the exo camera is set up to focus more on the working desk, while capturing user's details (hands, torso). Here, we often use the Sony APSC camera with a 30mm lens. For tasks such as assembly/disassembly of mechanical setups, the exo camera was set up to capture the entire room/lab (camera with a wider FOV). We also decided against setting up exo views in certain environments or cases, since our intention was to not inadvertently film others in the background without prior consent. For collaborative setups, there was also an additional challenge, since multiple workers can block the direct line between the camera and the device. We added a second exo view with a complementary perspective to address such cases. This needs further study, especially for real-world adoption.

## E.3  Egocentric and Exocentric Views for Understanding Industrial Processes

The impact of the egocentric, exocentric, and joint (ego + exo) views is task dependent. For example, the exo-view helps with the tasks where the task was not completely visible from the ego view or the mistake was not in focus. In certain other cases, the exo view did not improve the prediction and led to an incorrect prediction (based on a review of the zero-shot results in Table 5).

| Device 1: | Sony Alpha 6400 |
|---|---|
| Sensor Type | APS-C, CMOS |
| #Pixels | 24.2 megapixels |
| Video Resolution | $1920 \times 1080$ |
| Frame rate | 30FPS |
| Creative Style | Standard |
| Lens 1 | Sigma 16mm f/1.4 |
| Lens 2 | Sigma 30mm f/1.4 |
| Frame rate | 30FPS |
| Microphone | ON |
| **Device 2:** | **Samsung Galaxy A51** |
| Sensor 1 | 48MP f/2.0 (Standard) |
| Sensor 2 | 12MP f/2.2 (Ultra-wide) |
| Resolution | $1920 \times 1080$ |
| Microphone | ON |

Table 10: Details of the devices used for exocentric recordings, along with the accessories and custom settings.

## E.4  Synchronisation

The data from the exocentric cameras was recorded independently of the Aria device, which means that they are not automatically synchronised and calibrated w.r.t. the Aria device and the resulting 3D point cloud. For time synchronisation, we start all recordings at approximately the same time, and

trim the processed videos and frames later. This does not result in a guaranteed millisecond accurate synchronisation, like the Nymeria dataset [51] or Ego-Exo4D [14]. Similarly, spatial synchronisation and calibration between the Aria device and the other devices needs further work.

Since we had to move our setup around the facility, incorporating the manual synchronisation approach served as a practical solution. We rechecked the synchronisation data for the ego-exo recordings. But, we acknowledge, that the millisecond-level temporal details may be misaligned. We found the approach to be highly robust for the phenomena cases in our paper. Our benchmarks operate at the level of actions and task steps, which typically span multiple seconds. For these analyses, a sub-second alignment is sufficient and does not impact the validity of the results for tasks like mistake detection. We can add a short analysis on this if needed.

### E.5 Workstation for Data Processing and Experiments

We use a dedicated system for storing all the collected data, for processing and for most experiments. Table 11 gives the technical details.

| Parameter | Value |
|---|---|
| System Memory | 48 GB |
| CPU Cores | 12 |
| GPU Count | 1 |
| GPU type | NVIDIA RTX A6000 |
| Python version | 3.12.2 |
| PyTorch | 2.4.0 + cu121 |

Table 11: Details of the Workstation used for storing and processing the data.

## F  Annotations, Modalities, and Data Structure

This section describes the annotations, data modalities, and structural conventions used in the dataset to facilitate consistent interpretation and use.

### F.1  Annotation Schema and Modalities

The annotation protocol involves participants labelling their own videos. The decision was made to have the participants annotate their own data to balance the workload for the group. Secondly, we believe the participants are best suited to describe their actions and intentions, and do a thorough review of the work done, first-hand. A clear annotation protocol was established, and the participants were provided examples and templates for annotation. We would differentiate the annotations into two categories:

*Keysteps (and Mistake/Correct) Annotations:* These are objective, well defined, and are task-dependent. We provide additional details in Sections K and M in the Appendix. We also have clear labels and conventions (Supplementary Material) for the parts of a device/assembly and dependencies between the keysteps. Similarly, we established clear guidelines on what would constitute as a mistake, and the edge cases were discussed before annotation.

*Finegrained Actions:* These describe the short actions as verb + noun (and adjective) pairings, and are user dependent (for example, in order to remove bolts from a device, Participant 1 might loosen bolts before sorting them, and Participant 2 might remove each bolt and store it before moving to the next). These tend to be more subjective. We would argue that this reflects the natural diversity of the participant group and the way in which they might refer to an action, tool or their surrounding.

We are publishing the raw data from all sensors. Additionally, low resolution (480p) MP4 videos are also shared, along with full resolution videos (2880*2880). We obtained consent from Meta's Reality Labs for publishing the Machine Perception Service (MPS) output of the dataset. This includes the eye gaze estimation, hand pose, 3D semi-dense point cloud, as well as user trajectory. These can be used for other research investigations. For the MPS output, we use the default format and structure.

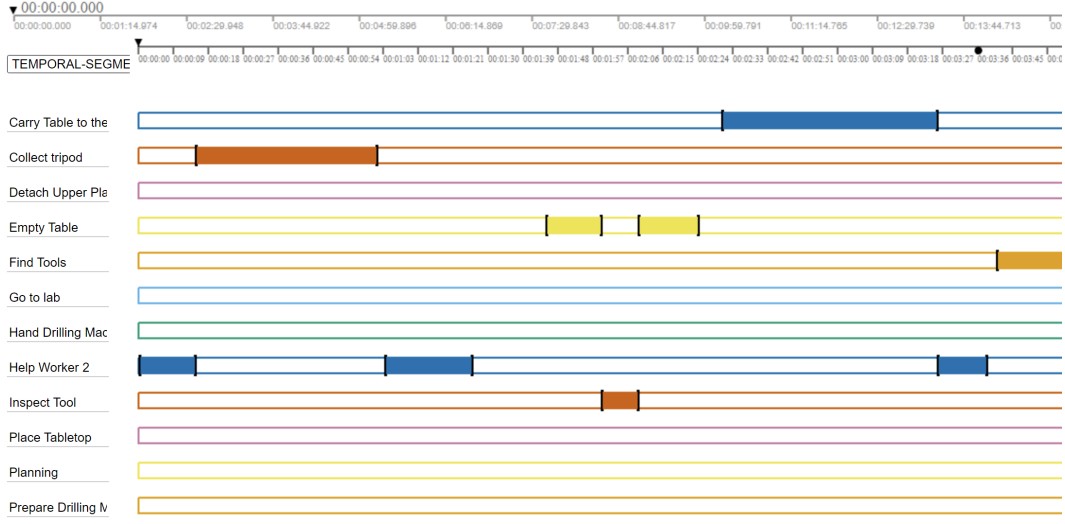

Figure 10: Example annotation for the *Disassembly* task from the perspective of Worker 1. The scenario has been shown in Figure 2 in the main paper and Figure 7. The actions include task specific labels such as *empty table, detach upper plate*, as well as collaborative actions, such as *Help Worker 2, Planning and Hand Drilling Machine to Coworker*. The action segments last between 2 seconds and 45 seconds.

The processed outputs from MPS are arranged in the default folder structure as explained in the Project Aria Docs [11].

## F.2 Naming Convention and Dataset Structure

All data files in the IndEgo dataset (including raw VRS recordings, MP4 videos, MPS outputs, and annotation files) share a unique, consistent filename. The naming convention encodes the scenario, user, location, task, and recording sequence.

The general format is:

$$[Scenario]\_([Type])\_[User]\_[Location]\_[Task]\_[Index]$$

The components are defined as follows:

- **Scenario ID (First 2 digits):** Indicates the broad scenario category:
    - 01: Assembly and Disassembly
    - 02: Inspection and Repair
    - 03: Logistics and Organisation
    - 04: Woodworking
    - 05: Miscellaneous
- **Type (Scenario 01 only):** For Scenario 01, an additional letter distinguishes the operation: a denotes *Assembly* and d denotes *Disassembly*.
    - *Example:* 01_d_... (Disassembly) vs. 01_a_... (Assembly).
- **User ID** (Uxx): Identifies the participant(s).
    - **Single User:** Uxx denotes the user capturing the data (e.g., U15).
    - **Collaborative Tasks:** For two-person tasks, the format is Uxx_Uyy. The first ID (Uxx) is the **camera wearer** (egocentric view), and the second ID (Uyy) is the **collaborator**.
    - *Note:* Collaborative sessions are recorded from both perspectives. For every file ...U02_U14... (User 2's view), there is a corresponding file ...U14_U02... (User 14's view).

- **Location ID** (Rxx)**:** Denotes the room or lab location (e.g., R03). If a task spans multiple areas, this indicates the primary location or starting point.
- **Task ID** (xx)**:** A unique identifier for the specific object or procedure within the scenario (e.g., 01 might correspond to a specific engine block setup).
- **Recording Index (Final digit):** Indicates the chronological order of recordings for a specific setup (e.g., 1, 2, 3).
  - *Example:* In Scenario 01, index 1 often corresponds to the initial disassembly, while index 2 corresponds to the subsequent re-assembly of the same object by the same user.

**Examples:**

- 04_u15_r03_01_1: A Woodworking task (04) performed by User 15 in Room 3, Task 01, first recording.
- 04_u15_u01_r04_01_1: A collaborative Woodworking task viewed from User 15's perspective, working with User 01.
- 01_d_u02_r03_01_1: A Disassembly task (01_d) by User 02 in Room 3.
- 01_a_u02_r03_01_2: The corresponding Assembly task (01_a) performed immediately after the disassembly above.

**Mistake Detection Naming Convention**    The Mistake Detection subset (covering 25 specific tasks) follows a slightly modified convention. The general format is:

$$[\text{Task}]\_([\text{Tool/Role}])\_[\text{User}]\_[\text{Location}]\_[\text{Condition}]\_[\text{Index}]$$

The components are defined as follows:

- **Task ID (First 2 digits):** Unlike the general dataset, the leading digits here correspond to the specific Mistake Detection task number (ranging from 01 to 25), rather than the broad scenario category.
- **Tool / Role Identifier (Optional):** For tasks that can be performed with different tools or roles, an identifier is inserted after the Task ID.
  - *Example:* sd denotes *screwdriver* and ed denotes *electric drill* (e.g., 17_sd_... vs. 17_ed_...).
  - Full details on task-specific tools and roles are provided in the guide on the project website.
- **User ID** (Uxx) **& Location ID** (Rxx)**:** These follow the same convention as the main dataset (see above). Collaborative tasks (Uxx_Uyy) also follow the logic where the first ID denotes the camera wearer.
- **Condition** (c / m)**:** Indicates the presence of errors in the sequence.
  - c: **Correct** execution (no mistakes).
  - m: **Mistake** execution (contains at least one error).
- **Recording Index (Final digits):** A chronological counter for the recordings of a specific setup (e.g., 01, 02).

**Examples:**

- 01_u01_u15_r05_c_01: Task 01, performed by User 01 (collaborating with User 15) in Room 5. The execution is correct (c), recording 01.
- 13_u17_r04_m_01: Task 13 by User 17 in Room 4, containing mistakes (m).
- 17_ed_u14_r06_c_01: Task 17 performed using an electric drill (ed) by User 14, correct execution.

**Directory Organisation**    The dataset is organised hierarchically to facilitate easy navigation. The top-level directories correspond to the main scenarios (e.g., Assembly, Logistics) and specific benchmark tasks and context (Tools, Singular Actions).

```
0_Datasheet_and_readme
1_Assembly
|-- 01
|   |-- Raw_Data
|   |   |-- Worker_1_Files
|   |   |   {VRS, MP4, Frames, Annotations,
|   |   |   Summary, Motion}
|   |   |-- Worker_2_Files
|   |   |   {VRS, MP4, Frames, Annotations,
|   |   |   Summary, Motion}
|   |   |-- Exo_Data
|   |   |   {MP4, Frames}
|   |   |-- Task_graph_and_steps
|   |-- MPS_Output
|   |-- Benchmark_Task_Data_and_References
|-- 02
...
1_Disassembly
|-- ...
...
2_Inspection_Repair
|-- ...
...
6_Tools_Objects_in_Context
|-- 01_Drilling_Machine
|   {VRS, MP4, MPS_Output}
...
7_Tools_Objects_demo
|-- 01_Drilling_Machine
|   {VRS, MP4, MPS_Output}
...
|-- Tools_List
8_Singular_Actions
|-- 01_Fastening_a_Bolt
|   {VRS, MP4, MPS_Output}
...
|-- Actions_List
|-- Benchmark_tasks_and_references
9_VPR
Mistake_Detection
|-- 01_Set_up_camera_and_tripod
|   |-- Raw_Data
|   |-- {0, 1, 0, 0, 0, 1, 0, 0} //Label
...
```

# G   Dataset and Code Availability

Our dataset is large and multimodal, consisting of egocentric and exocentric videos, audio, eye gaze, SLAM, and action annotations across a wide range of industrial tasks. The dataset is available on Hugging Face Hub[3] for review.

The accompanying code and scripts are provided via our GitHub repository[4]. This includes:

- Scripts for parsing, preprocessing, and sampling egocentric data (from Project Aria Tools).

- An exploration-focused Collab notebook for loading and visualising data.

---

[3] https://huggingface.co/datasets/FraunhoferIPK/IndEgo/
[4] https://github.com/Vivek9Chavan/IndEgo/

| Dataset | Scenario | Focus |
|---|---|---|
| EgoExo4d [14] | Skilled physical activities across 8 tasks | Cross-view Representation Learning, Proficiency Estimation |
| Aria Everyday Activities [42] | Everyday Activities | General multimodal AI and egocentric vision research on day-to-day tasks in typical environments. |
| Aria Digital Twin [48] | Daily & Indoor | AR/VR applications. Real-world with Digital Twin and Dynamic, photorealistic digital counterpart alongside real-world data. |
| Aria Synthetic Environments [52] | Synthetic. Diverse & Indoor | 3D scene understanding. Procedurally generated, large-scale dataset for ML training. |
| Digital Twin Catalogue | Object Reconstruction | High-quality dataset for detailed object reconstruction research. |
| Aria Everyday Objects | Daily, Indoor | Small-scale, Annotated and High-quality 3D Oriented Bounding Box (OBB) annotations in a real-world context. |
| HOT3D [32] | Indoor, Kitchen, Desk | Hand-Object Interaction |
| Nymeria [51] | Diverse | Human motion understanding |
| IndEgo (Ours) | Industrial Settings & Tasks | Collaborative Task Understanding, Mistake Detection, Reasoning and Planning, Egocentric Assistants |

Table 12: Comparison of IndEgo dataset with other publicly available datasets collected using the Project Aria toolkit [4]. For brevity, we only focus on limited aspects of each dataset. Our dataset is unique in terms of its setting and focus.

- Benchmarking baselines and evaluation scripts for the tasks described in this paper.

Several components borrow from existing open-source libraries (mostly under Apache 2.0 or MIT License), which are credited and documented accordingly. Instructions for reproducing the benchmark experiments are included, and additional utilities for extending the dataset will be released progressively.

**Maintenance and Future Plans.** The dataset and its associated resources will be actively maintained by the authors and the research teams at Fraunhofer IPK and TU Berlin. We are committed to the long-term value of this benchmark. Should any inaccuracies in the annotations be found, we will periodically publish updated versions. Furthermore, we plan to expand the dataset in the future with additional tasks, scenarios, and useful insights to continue driving research in this domain. We welcome community feedback and contributions via the Hugging Face and GitHub repositories.

## H   Comparison with Other Datasets

Table 12 summarises the scenario and focus of other publicly available datasets that used the Aria device for data collection, and compares them with the IndEgo dataset.

Compared to prior egocentric video datasets, our collection focuses on longer-duration and movement-intensive industrial tasks. In our subjective assessment, we observe that the performance of existing vision-language models tends to degrade on longer videos, particularly when summarisation or task inference is required over extended sequences. Our dataset includes tasks lasting up to 68 minutes, presenting a more realistic and challenging setting for egocentric video understanding.

Additionally, our dataset uniquely covers logistics, manual handling, and other labour-intensive industrial scenarios that are often underrepresented in existing benchmarks. These tasks introduce increased motion, scene variation, and object manipulation complexity, which further challenge models in summarisation, action recognition, and collaborative reasoning. By addressing these

gaps, our dataset pushes toward more practical and robust egocentric AI systems in real-world environments.

# I  Details on Experiments

Table 13 gives the details of the setup for the zero-shot and finetuning experiments. We use the same setup for all our ML trainings for a fair and unbiased comparison. For the finetuning experiments, we use an 80/20 split (Mistake Detection, Action Recognition).

| Parameter | Value |
|---|---|
| Temperature | 0.2 |
| Input Resolution (Video) | (480, 480) |
| Max. Frames | 16 |
| Top-p | 0.95 |
| Max. Output Tokens | 512 |
| Input Resolution (Image) | (240, 240) |
| Batch Size | 64 |

Table 13: Hyperparameter settings used for the zero-shot and finetuning evaluations with SOTA Video Language Models (VLMs).

## I.1  Reasoning-Based Video Question Answering

We use a common set of input prompts to nudge the model to understand the task and answer the question. The answers generated by the VLM are judged by Mistral-Large2 [62]. The standard prompt for the two models is given below.

```
To VLM: You are watching an egocentric recording captured by a user
performing a task.  Analyse the input video, the objects, the action, and
think carefully about the question.  Select one of the options (or two if
both are correct) and explain your choice in short.
Input:  Video, Question, Options (a, b, c, d, e)
```

```
To LLM: You are analysing the answer given by a VLM on a reasoning based
question.  Carefully review the question, the option, the correct answer,
the action label, the VLM's response and its reasoning.  Decide whether
the VLM answered the question correctly, with correct reasoning.  If yes
award it 1 point.  If there are two correct answers and the VLM selects one,
award it 0.5 point.  If the VLM selects an incorrect answer, award it 0
points.
Input:  Task label, Question, Options, Correct Answer, VLM response
```

## I.2  Mistake Detection

For zero-shot evaluation of the mistake detection benchmark, the models were prompted to see whether the *corrrect* steps are being performed and whether the *erroneous* steps are being avoided. Following are the prompts for checking both. Variations of the base prompts are generated by Mistral for added prompting. The VLM is given small video segments separately for processing, along with the two types of prompts. The VLM is consulted with both prompt types multiple times, and the average response is taken.

```
To VLM: You are watching an egocentric recording captured by a user
performing a task.  Analyse the input video, the objects, the action, and
think carefully whether the user performed the correct action.
Input:  Video
Task:  Open hatch of the trolley and load objects
Expected:  The user opens the hatch on both ends, lowers the door and loads
objects.
Questions:  Did the open the hatch before loading objects?  Did the user
open the hatch from both sides?  Was the hatch open when the user loaded
the parts?
```

```
To VLM: You are watching an egocentric recording captured by a user
performing a task.  Analyse the input video, the objects, the action, and
think carefully whether the user performed the correct action.
Input:  Video
Task:  Open hatch of the trolley and load objects
Common mistakes:  The user opens forgets to open the hatch.  The user opens
the hatch on only one side
Questions:  Did the user not open the hatch?  Was the hatch on the door
still open at the end of the video?  Did the user load objects onto the
trolley without opening the hatch?
```

## I.3   Task Understanding in a Collaborative Setting

We assess the zero-shot capabilities of the SOTA VLMs to assess their ability to understand collaborative actions and tasks. For this, we input small video segments, similar to *Mistake Detection* and query the VLMs to check whether they understand which user performed which action. A typical example of the question asked is shown below. Green denotes VLMs answer this correctly (on average), Red denotes VLMs struggle to answer correctly. The models often identify the general context. However, they cannot understand how the action of one user differs from the other user reliably. Moreover, they often fail to appropriately follow and remember which the person and their actions, esp. when the two users work closely and their hands are in proximity to each other.

```
To VLM: Here is an egocentric view of a task I am performing with my
coworker.  What are we working on?  What are our roles relative to each
other?
```

```
To VLM: Here is an egocentric view of a task I am performing with my
coworker.  I am wearing the black shirt, and the coworker has the grey
hoodie.  What is the other person doing?  How is it different from what I
am doing?
```

```
To VLM: Here is an egocentric view of a task I am performing with my
coworker.  I am wearing the black shirt, and the coworker has the grey
hoodie.  Here is a step by step description of the task.  Answer which of
these is performed by me and which is performed by the coworker.
Steps:  Hold the frame and the cover together, fasten the bolt, inspect the
assembly
```

## I.4   Action Recognition

We use a similar setup as for fine-tuned MD evaluation for the top 100 actions (verbs) in the dataset, reporting an accuracy between 57.6% (QVL + MLP) and 64.1% (VL3 + Tr). Even for short and well-defined actions, we observe frequent misclassification between semantically opposite actions such as *attach* and *detach*, or *open* and *close*. These errors highlight the importance of temporal reasoning, as such actions may appear visually similar in isolated frames but differ in their temporal progression and causal context. Furthermore, the egocentric perspective presents additional challenges due to rapid head motion, frequent occlusions from hands or tools, and narrow fields of view. These factors contribute to ambiguity in visual cues and limit the effectiveness of spatial-only models, underscoring the need for models that incorporate both temporal context and egocentric motion dynamics for accurate action recognition. Prior works have also highlighted this [13].

## J  Reasoning-Based QA

We use LLMs/VLMs to generate potential ideas and proposals for the questions, which are then reviewed and edited by the team. The question and answer choices are chosen to keep the visual and contextual information about the video data and the action vague. Each question is linked to a specific recording in the dataset, and cannot be independently answered without the visual and reasoning background. Some examples of the question types include: scene understanding, reasoning what should come before or after the task, understanding whether the task is vital in the broader context of a larger sequence (e.g. assembly), possible solutions to problems, among others.

```
Action:  Disconnecting a drilling machine after use

Question:  The person who was performing this action was injured.  What was
likely the reason for this?

a.  Sharpness of the tool
b.  Poor electrical insulation
c.  The weight of the machine, along with the poor weight distribution
d.  The miscommunication with other colleagues
e.  There is no likelihood of an injury with this task
```

```
Action:  Wearing Safety Shoes

Question:    When is this action likely to occur?

a.  When dealing with a sharp object
b.  When dealing with a heavy object
c.  When the user plans on walking a long distance
d.  After the user has finished work
e.  This action is not likely to occur, unless there is an unusual
circumstance
```

## K  Mistake Detection

In this section, we describe the mistake detection benchmark of the dataset.

### K.1  Intentional and Unintentional Mistakes

The participants were first briefed on the correct sequence of steps involved in each task. To create a robust benchmark, some mistakes were deliberately planned and recorded. These planned mistakes were selected after discussing likely errors with participants and domain experts, such as skipping a safety step, using the wrong tool, or performing actions in the wrong order. This approach allowed us to capture a broad and meaningful range of mistake types, which would be unlikely to occur frequently through natural recording alone.

Participants then recorded both correct and incorrect executions of the same tasks. Table 14 outlines the tasks, correct step sequences, and associated common mistakes for various activities in the IndEgo dataset. The mistake detection tasks and the steps were predefined, aimed at covering different scenarios. However, the participants were not controlled in a stringent manner. They were asked to perform the activities repetitively, just as they would in a real application. Several mistakes captured in the dataset are unintentional, i.e. the participant made a mistake when they did not mean to (including fatigue-related errors). Data collection and development of the benchmark was an iterative process, where we expanded the list of mistakes based on the data. In order to balance the data and diversity, we then added more correct recordings if needed. The decision to have a list of predefined mistakes at the start was made to include diverse mistakes, and failure scenarios, without having to wait for the particular mistake to occur. As the results show, there is a wide gap between the performance of current SOTA models, and the requirements for robust mistake detection from ego/exo data. We are happy to include this in the appendix. We believe our work will highlight the need for further research and development in this area.

Figure 11 shows a distribution of the egocentric recordings based on their duration (in seconds) and the application scenarios. Approximately 38% of the recordings are *correct*, with no mistakes, and 62% of the recordings contain at least one *mistake*.

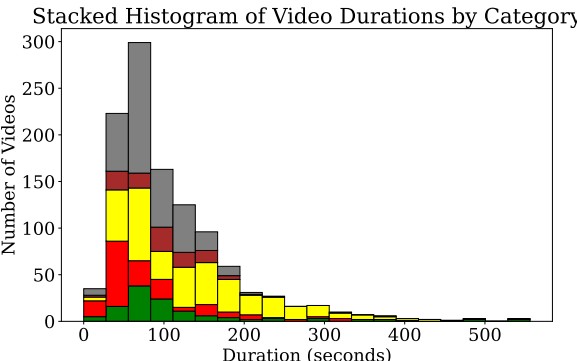

Figure 11: Stacked histogram of video durations (sec) by category for *Mistake Detection*: assembly/disassembly , inspection/repair , logistics/organisation , woodworking , and miscellaneous .

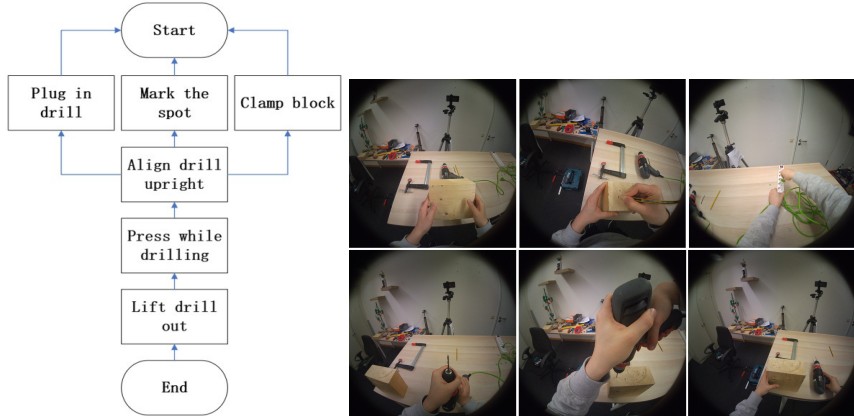

Figure 12: **Top:** Task graph for Task 8 (*woodworking, drilling a hole*) of the Mistake Detection data. The sequence of actions is from the top to the bottom, and the dependencies are shown with an arrow. **Bottom:** Frames from a sample recording from the egocentric perspective for the task.

**Mistake Categorization.** The possible mistakes in the 25 industrial tasks of the mistake detection portion of IndEgo are categorised into four types: severe mistakes, mistakes causing process failure, mistakes that impact future steps, and mistakes that can cause harm. While some mistakes fit neatly into a single category, others can belong to multiple categories at once.

A severe mistake is one that compromises the integrity of the task or results in significant disruptions. For example, in the task of loading a trolley, failing to place the items securely could cause them to fall over during transport. This could lead not only to damage of the fallen object, but also others loaded alongside it. A mistake causing complete process failure prevents the task from being successfully completed. Forgetting to load one of the required items onto the trolley would mean that the proper steps were not followed, the task is incomplete and must be redone.

A mistake that impacts future steps does not immediately halt the process but can cause complications later on or make other steps invalid. For instance, neglecting to open the hatch before loading the trolley makes loading it a lot more difficult, but also negates the necessity of the step of closing the hatch.

Finally, a mistake that can cause harm poses a risk to worker safety. An example of this is not wearing protective gloves when handling heavy or sharp objects, while loading them onto the trolley, thereby possibly leading to injuries. Such categorisation is essential, since two visually similar mistakes can have varying significance, e.g. not wearing gloves before a task is much worse than forgetting to remove gloves after the operation.

| Task 1 | Scenario | Correct Steps | Common Mistakes |
|---|---|---|---|
| Set up camera and tripod | Miscellaneous | 1. Bring tripod to marked spot 
 2. Level tripod 
 3. Fasten tripod legs 
 4. Attach plate to camera 

 5. Mount camera on tripod 
 6. Level and fasten tripod head 
 *Steps 4+5 and 6 are interchangeable.* | Setting tripod at the wrong spot 
 Skipping step 
 Skipping step 
 Skipping step, attaching loosely, attaching wrong way 
 Skipping step, insecure mount 
 Skipping step |

| Task 2 | Scenario | Correct Steps | Common Mistakes |
|---|---|---|---|
| Packaging objects in a box | Logistics | 1. Wrap fragile item in bubble wrap 
 2. Place objects in the box 


 3. Close and seal the box with tape 
 *Steps 1 and 2 are interchangeable.* | Skipping step 
 Skipping step, forgetting object, placing heavy objects on top of fragile ones, placing objects so that the box does not close 
 Skipping step, taping wrong |

**Annotations.** In annotating the recordings of the mistake detection part, we followed a step annotation approach, using standardised steps for each task and giving the start and end time stamps, between which these are performed in the videos. Additionally, each video is categorised with either a "c" or an "m" in the name to indicate whether it is a correct or mistake recording. For each step in each video, we also noted whether a mistake was being made and, if so, what kind of mistake occurred, or if the step was performed at all.

### K.2 Mistakes in Longer Action Sequences

These mistakes occur in the longer task sequences of the dataset. Common mistakes involve choosing an inappropriate working desk, wrong-sized tool, incorrect reasoning and actions for an *unguided* task, etc.

## L Summarisation

Summarising long egocentric videos into concise, multimodal descriptions is a key challenge in making wearable AI systems useful and efficient. In industrial settings, workers often engage in extended sequences of actions across multiple stages of a task. Raw video is too dense for downstream consumption, and structured summarisation enables better indexing, querying, and review of workflows. Summarisation can also support retrospective training, documentation, and performance evaluation.

Our approach segments egocentric videos into fixed one-minute intervals and prompts a vision-language model (VLM) to generate a concise noun-verb description for each segment. These summaries are designed to capture the primary action(s) in the scene without requiring explicit

| Task 3 | Scenario | Correct Steps | Common Mistakes |
|---|---|---|---|
| Putting on a safety kit | Miscellaneous | 1. Put on safety jacket and close 
 2. Put on safety shoes and tighten 
 3. Put on safety gloves 
 4. Put on ear muffs 
 *All steps are interchangeable.* | Skipping step, not closing 
 Skipping step, not tightening 
 Skipping step, only one glove 
 Skipping step, not covering ears |

| Task 4 | Scenario | Correct Steps | Common Mistakes |
|---|---|---|---|
| Replacing a box cutter blade | Miscellaneous | 1. Put on safety gloves
2. Remove the blade and slider

3. Attach slider to new blade

4. Insert new blade

*Steps 3 and 4 are interchangeable.* | Skipping step
Pushing blade up instead of retracting
Skipping step, attaching the wrong way
Skipping step, inserting the wrong way, inserting old blade |

| Task 5 | Scenario | Correct Steps | Common Mistakes |
|---|---|---|---|
| Changing a light bulb | Inspection & Repair | 1. Unplug lamp
2. Remove old bulb
3. Insert new bulb

4. Plug in lamp
5. Test lamp | Skipping step
Skipping step
Skipping step, inserting old bulb, inserting loosely
Skipping step, plugging in too early
Skipping step |

| Task 6 | Scenario | Correct Steps | Common Mistakes |
|---|---|---|---|
| Leaving a work station/room | Logistics and Organisation | 1. Turn off and unplug equipment
2. Put away all items
3. Clean work station
4. Close windows
5. Turn off lights
6. Close door
*Steps 1-4 are interchangeable.* | Skipping step
Skipping step, forgetting item
Skipping step
Skipping step
Skipping step
Skipping step |

| Task 7 | Scenario | Correct Steps | Common Mistakes |
|---|---|---|---|
| Changing a drill bit or a screw bit | Miscellaneous | 1. Turn off/unplug drill/put in neutral
2. Remove old bit
3. Insert new bit


4. Turn on/plug in drill/put in non-neutral
5. Test drill | Skipping step
Skipping step
Skipping step, inserting old bit, insert bit between two jaws, not tightening chuck
Skipping step

Skipping step |

| Task 8 | Scenario | Correct Steps | Common Mistakes |
|---|---|---|---|
| Drilling a hole into a wooden block | Woodworking | 1. Mark spot to drill with pencil
2. Clamp block to table
3. Plug in drill
4. Drill hole


*Steps 1-3 are interchangeable.* | Skipping step
Skipping step, clamping loosely
Skipping step
Skipping step, drilling at an angle, not applying pressure, drilling at wrong spot, breaking bit off |

| Task 9 | Scenario | Correct Steps | Common Mistakes |
|---|---|---|---|
| Screwing in a bolt with an electric drill | Assembly-Disassembly | 1. Change screw bit

2. Plug in drill
3. Screw in bolt | Skipping step, wrong screw bit, forgetting screw bit
Skipping step
Skipping step, screwing at angle, screwing in wrong direction |

| Task 10 | Scenario | Correct Steps | Common Mistakes |
|---|---|---|---|
| Loading objects into a big trolley | Logistics and Organisation | 1. Open hatch
2. Put on safety gloves
3. Load objects into trolley

4. Close hatch

5. Test trolley
*Steps 1 and 2 are interchangeable.* | Skipping step
Skipping step
Skipping step, not loading securely, forgetting object
Skipping step, closing only one latch
Skipping step |

| Task 11 | Scenario | Correct Steps | Common Mistakes |
|---|---|---|---|
| Starting a PC | Inspection and Repair | 1. Attach power cable
2. Plug in power cable
3. Attach display cable
4. Plug in display
5. Turn on power
6. Attach keyboard
7. Attach mouse
8. Turn on pc

*All steps are interchangeable.* | Skipping step, attaching loosely
Skipping step
Skipping step
Skipping step
Skipping step
Skipping step
Skipping step
Skipping step |

| Task 12 | Scenario | Correct Steps | Common Mistakes |
|---|---|---|---|
| Weighing objects | Miscellaneous | 1. Turn on scale
(2. Place box)
3. Tare
4. Weigh object

5. Note down weight
6. Remove object
7. Turn off scale
*Steps 2-6 are repeated for every object.* | Skipping step, non-empty box
Skipping step
Skipping step, object halfway off the scale, hand on scale
Skipping step
Skipping step
Skipping step |

| Task 13 | Scenario | Correct Steps | Common Mistakes |
|---|---|---|---|
| Putting away an electric drill | Miscellaneous | 1. Take off battery
2. Plug in charger
3. Charge battery
4. Remove screw bit and extension

5. Store screw bit and extension
6. Put drill and bit case in case

7. Close case
*Steps 1-3 and 4-6 are interchangeable.* | Skipping step
Skipping step
Skipping step
Skipping step, removing only one part
Skipping step, forgetting either
Skipping step, forgetting either, adding item
Skipping step |

| Task 14 | Scenario | Correct Steps | Common Mistakes |
|---|---|---|---|
| Re-organizing a tool box | Logistics and Organisation | 1. Place all tools securely

2. Secure middle divider
3. Close tool box | Skipping step, forgetting tool, adding other items, placing loosely, wrong spot
Skipping step
Skipping step, not closing securely |

| Task 15 | Scenario | Correct Steps | Common Mistakes |
|---|---|---|---|
| Taking a picture on a multi view setup | Miscellaneous | 1. Turn on light | Skipping step |
| | | 2. Attach lens | Skipping step |
| | | 3. Attach plate | Skipping step, attaching wrong way |
| | | 4. Take lens cover off | Skipping step |
| | | 5. Mount on tripod | Skipping step |
| | | 6. Turn on camera | Skipping step |
| | | 7. Take picture | Skipping step, not centering object, done four times |
| | | 8. Rotate Object by 120° | Skipping step, rotating in wrong direction, done four times |
| | | *Steps 1-4 are interchangeable and steps 7+8 are repeated three times in total.* | |

| Task 16 | Scenario | Correct Steps | Common Mistakes |
|---|---|---|---|
| Changing batteries | Inspection and Repair | 1. Test appliance | Skipping step |
| | | 2. Take out old batteries | Skipping step, taking out only one |
| | | 3. Insert new batteries | Skipping step, inserting only one, inserting old ones, wrong size, wrong way |
| | | 4. Close appliance | Skipping step |
| | | 5. Test appliance | Skipping step |

| Task 17 | Scenario | Correct Steps | Common Mistakes |
|---|---|---|---|
| Creating a butt joint | Woodworking | 1. Put on safety gloves | Skipping step |
| | | 2. Clamp piece 1 | Skipping step |
| | | 3. Place piece 2 | Skipping step, aligning piece wrong |
| | | 4. Screw in screws | Skipping step, only one screw, using hands, using hammer, screws loose |
| | | 5. Unclamp butt joint | Skipping step |
| | | 6. Inspect butt joint | Skipping step |
| | | 7. Return clamp to original position | Skipping step |
| | | 8. Take off safety gloves | Skipping step |

| Task 18 | Scenario | Correct Steps | Common Mistakes |
|---|---|---|---|
| Transporting heavy palette with a pallet jack | Logistics and Organisation | 1. Put on safety shoes | Skipping step |
| | | 2. Put on safety gloves | Skipping step |
| | | 3. Pick up package | Skipping step, picking up by hand |
| | | 4. Bring to destination | Skipping step, wrong spot, wrong destination, bringing by hand |
| | | 5. Help colleague | Skipping step |
| | | 6. Return jack | Skipping step, leaving jack midway, wrong jack |
| | | 7. Take off safety gloves | Skipping step |
| | | 8. Take off safety shoes | Skipping step |
| | | *Steps 1-2 and 7-8 are interchangeable.* | |

| Task 19 | Scenario | Correct Steps | Common Mistakes |
|---|---|---|---|
| Preparing a shipment for transportation | Logistics and Organisation | 1. Put on gloves
2. Place package on palette
3. Tape the package shut

4. Wrap the package with foil

5. Secure the package with belts

6. Label the package
*Steps 2 can be done after 3 or 4.* | Skipping step
Skipping step, not placing in center
Skipping step, taping halfway, taping wrong
Skipping step, wrapping halfway, wrapping only one side
Skipping step, only using one belt, belts loose
Skipping step, label loose |

| Task 20 | Scenario | Correct Steps | Common Mistakes |
|---|---|---|---|
| Transport heavy objects between floors | Logistics and Organisation | 1. Put on gloves
2. Put in brakes
3. Raise trolley platform
4. Load devices onto trolley
5. Lower trolley platform
6. Take elevator
7. Unload devices from trolley

8. Park trolley
*Step 2 is repeated before every loading, unloading and taking the elevator, steps 3+5 are repeated before every loading and unloading.* | Skipping step
Skipping step, using hands
Skipping step, using hands
Skipping step, only loading one
Skipping step
Skipping step, wrong floor
Skipping step, only unloading one, wrong spot, taking detours
Skipping step |

| Task 21 | Scenario | Correct Steps | Common Mistakes |
|---|---|---|---|
| Transporting items across rooms | Logistics and Organisation | 1. Put on gloves
2. Load the two objects from room 1 onto the trolley

3. Load the two objects from room 2 onto the trolley

4. Load the two objects from room 3 onto the trolley

5. Load the two objects from room 4 onto the trolley

6. Load the two objects from room 5 onto the trolley

7. Unload all objects from the trolley near the elevator | Skipping step
Skipping step, forgetting object, throwing object to colleague, not loading securely, not loading first
Skipping step, forgetting object, throwing object to colleague, not loading securely, not loading second
Skipping step, forgetting object, throwing object to colleague, not loading securely, not loading third
Skipping step, forgetting object, throwing object to colleague, not loading securely, not loading fourth
Skipping step, forgetting object, throwing object to colleague, not loading securely, not loading last
Skipping step, not unloading all objects |

| Task 22 | Scenario | Correct Steps | Common Mistakes |
|---|---|---|---|
| Repairing a dysfunctional PC | Inspection and repair | 1. Open the pc
2. Attach the battery
3. Attach the RAM chip
4. Attach the hard disk

5. Attach the fan
6. Attach the power cable
7. Clean the inside

8. Close the pc
9. Test the pc
*Steps 2-7 are interchangeable.* | Skipping step, attaching loosely
Skipping step, attaching loosely
Skipping step, attaching loosely, attaching only one cable
Skipping step, attaching loosely
Skipping step, attaching loosely
Skipping step, leaving the cloth inside, using hands
Skipping step
Skipping step |

| Task 23 | Scenario | Correct Steps | Common Mistakes |
|---|---|---|---|
| Assembling a camera stand | Assembly-Disassembly | 1. Attach vertical bar
2. Attach holder bar

3. Attach camera

4. Inspect assembly
*Steps 1-3 are interchangeable.* | Skipping step, attaching loosely
Skipping step, attaching loosely, wrong side
Skipping step, attaching loosely, wrong way
Skipping step |

| Task 24 | Scenario | Correct Steps | Common Mistakes |
|---|---|---|---|
| Disassembling a camera stand | Assembly-Disassembly | 1. Detach camera
2. Detach holder bar
3. Detach vertical bar
4. Sort parts
*Steps 1-3 are interchangeable.* | Skipping step, not completely
Skipping step, not completely
Skipping step, not completely
Skipping step |

| Task 25 | Scenario | Correct Steps | Common Mistakes |
|---|---|---|---|
| First aid for hand injury | Miscellaneous | 1. Notify colleague
2. Put pressure on wound

3. Put on gloves
4. Wash wound

5. Disinfect wound

6. Bandage wound
*Step 2 can be done multiple times.* | Skipping step
Skipping step, not helping, using hands, reacting late
Skipping step, putting on only one
Skipping step, not helping, washing twice, wrong order, scrubbing
Skipping step, wrong order, not helping, using hands, using old rag
Skipping step, not helping |

Table 14: Details on the tasks and scenarios from the *Planned Mistake Detection* benchmark of the IndEgo dataset, along with the *correct steps* and *common mistakes*.

temporal annotations. We use SOTA VLMs in a zero-shot setting, querying it with a minimal prompt that asks for concise action descriptions. The model processes sampled frames from each video segment and produces a text summary.

We additionally provide ground-truth annotations consisting of high-level action phrases for each minute of video. To evaluate model-generated summaries, we use Mistral-Large to determine whether the summarised output semantically aligns with the annotated ground truth. Each summary and corresponding ground truth are judged independently, and overall accuracy, precision, recall, and F1-score are computed across all segments.

This benchmark forms a foundation for more advanced summarisation techniques, such as hierarchical temporal summarisation, multimodal summarisation incorporating speech and gaze, and summarisation conditioned on downstream goals (e.g., safety auditing or skill verification). As wearable AI systems become more integrated into real-world workflows, effective summarisation will play a critical role in enabling scalable human-AI collaboration.

## M    Task Understanding in a Collaborative Setting

Understanding tasks from multiple egocentric viewpoints is essential for developing collaborative AI assistants in industrial environments. Collaborative understanding is critical in factories and workshops, where multiple operators jointly assemble, inspect, or repair devices. Unlike single-agent systems, collaborative scenarios require the AI to model not only what the user is doing, but also how they are coordinating with a coworker, potentially assuming roles such as teacher, learner, or equal collaborator. This becomes especially relevant in training, troubleshooting, or mixed-skill pairings where human interaction directly impacts task progression and outcomes. Given the economic and practical significance of collaboration, this benchmark introduces a novel and relevant research domain.

To benchmark task understanding in collaborative scenarios, we collected synchronised egocentric video recordings from two individuals jointly performing a task. Each person wore an egocentric device, and the videos were segmented into minute-long clips. For each segment, the model is expected to output the main action performed by the user, the main action of the coworker, and the respective roles each individual plays (teacher, student, or collaborator). This allows for a structured understanding of temporal coordination and implicit role shifts in collaborative tasks.

As a baseline, we use a zero-shot setup with different VLMs. The model receives both videos for each segment and a prompt instructing it to output a JSON-style response with the inferred actions and roles. This approach does not rely on fine-tuning or supervision and serves as a starting point for evaluating general-purpose VLMs on collaborative task understanding.

For evaluation, we provide manually annotated ground truth describing the actions and roles for each user across all segments. The model outputs are evaluated using Mistral-Large, which is prompted to determine whether the predicted fields match the ground truth semantically. Accuracy is computed across both users, and agreement is assessed per segment. We also conducted a small study (ca. 20% of the collaborative data) on the impact of different modalities on task understanding. The results show that removing audio modality impacts zero-shot performance (drop of 7%), especially since the coworkers often communicate with each other as they work together. Similarly, we observe that incorporating eye gaze can improve performance when the model is prompted to focus on the user's intention.

This setup opens promising directions for future research, including joint modelling of multi-agent temporal dynamics, role inference under noisy or ambiguous input, timeline alignment across viewpoints, and multimodal fusion across egocentric, exocentric, and environmental signals. As collaborative AI systems mature, understanding inter-human dynamics will be crucial for enabling intelligent assistance in real-world, multi-agent scenarios.

## N    Keysteps for Procedural and Non-Procedural Tasks

This section describes the keysteps for the medium to long task sequences in the dataset. **Keysteps** are coarse, semantically meaningful steps that represent the essential components of a procedural task. Rather than capturing fine-grained or atomic actions, keysteps correspond to higher-level operations

| No. | Steps | Dependencies |
|-----|-------|--------------|
| 01 | Preparation | — |
| 02 | Cut Zip Ties | — |
| 03 | Remove Wooden Plate | 02 |
| 04 | Remove Connector Plates | 03 |
| 05 | Remove Side Plates | — |
| 06 | Remove Power Connection | — |
| 07 | Remove Bottom Plate | 04 |
| 08 | Remove Top Plate | — |
| 09 | Remove Ram Chip | 08 |
| 10 | Remove Hard Disk Cables | 07 |
| 11 | Remove Hard Disk | 10 |
| *Note: Steps 05-08 and steps 09-10 are interchangeable.* | | |

Table 15: Keysteps and Dependencies for Disassembling Embedded Computing Unit

| No. | Steps | Dependencies |
|-----|-------|--------------|
| 01 | Preparation | — |
| 02 | Disassemble Upper Hinge | — |
| 03 | Disassemble Middle Hinge | — |
| 04 | Disassemble Lower Hinge | — |
| *Note: Steps 02-04 are interchangeable.* | | |

Table 16: Keysteps and Dependencies for Disassembling IKEA Lamp

such as "preparation," "load trolley," or "detach tabletop." These steps reflect the natural segmentation used by skilled workers when describing or performing tasks and are crucial for conveying the overall structure and intent behind a procedure.

In our framework, keysteps form the foundational units of **task graphs**. Each keystep is represented as a node in the graph, and directed edges between them encode the temporal or logical dependencies, i.e., which steps must be completed before others can begin. This representation enables reasoning about task flow, supports mistake detection when steps are skipped or performed out of order, and serves as a scaffold for instructional assistance and planning systems.

| No. | Steps | Dependencies |
|-----|-------|--------------|
| 01 | Preparation | — |
| 02 | Put on Gloves | — |
| 03 | Mark Tabletop | — |
| 04 | Remove Tabletop | 03 |
| 05 | Adjust Height | 04 |
| 06 | Remove Crank | — |
| 07 | Remove Lower Plate | — |
| 08 | Remove Upper Cover | 06, 07 |
| 09 | Separate Legs | 08 |
| 10 | Remove Feet | — |
| 11 | Clean up | — |
| *Note: For table without tabletop leave out steps 03-04.* | | |

Table 17: Keysteps and Dependencies for Disassembling Mechanical Table

| No. | Steps | Dependencies |
|-----|-------|--------------|
| 01 | Preparation | — |
| 02 | Put on Gloves | — |
| 03 | Remove Tabletop | — |
| 04 | Detach Hangers | 03 |
| 05 | Disassemble Upper Frame | 03 |
| 06 | Disconnect Center Bar | — |
| 07 | Disconnect Legs | — |
| 08 | Clean up | — |
| *Note: Steps 04-07 are interchangeable.* | | |

Table 18: Keysteps and Dependencies for Disassembling IKEA Table

| No. | Steps | Dependencies |
|-----|-------|--------------|
| 01 | Preparation | — |
| 02 | Put on Gloves | — |
| 03 | Remove Vertical Bar | — |
| 04 | Remove Plates | — |
| 05 | Detach Wheels | 03 |
| 06 | Remove Horizontal Bar | 04 |
| 07 | Disassemble Frame | 06 |
| 08 | Clean up | — |
| *Note: Steps 04-05 are interchangeable.* | | |

Table 19: Keysteps and Dependencies for Disassembling Trolley Prototype

| No. | Steps | Dependencies |
|-----|-------|--------------|
| 01 | Preparation | — |
| 02 | Put on Gloves | — |
| 03 | Open PC | — |
| 04 | Remove Chip Rack | 03 |
| 05 | Remove Duct | 03 |
| 06 | Remove Fan | 03 |
| 07 | Remove GPUs | 03 |
| 08 | Remove HDDs | 03 |
| 09 | Remove Ram Chips | 03 |
| 10 | Remove Heat Sink | 05 |
| 11 | Remove CMOS Battery | 07 |
| 12 | Remove CPU Frame | 10 |
| 13 | Remove CPU | 12 |
| 14 | Unplug Cables | 07 |
| 15 | Clean up | — |
| *Note: Steps 04-09 are interchangeable.* | | |

Table 20: Keysteps and Dependencies for Disassembling Workstation Computer

| No. | Steps | Dependencies |
|---|---|---|
| 01 | Preparation | — |
| 02 | Remove Camera and Plate | — |
| 03 | Remove Upper Bar | — |
| 04 | Remove Vertical Bar | — |
| 05 | Separate Base | — |
| 06 | Clean up | — |
| *Note: Steps 02-05 are interchangeable.* | | |

Table 21: Keysteps and Dependencies for Disassembling Small Camera Assembly

| No. | Steps | Dependencies |
|---|---|---|
| 01 | Preparation | — |
| 02 | Remove Vertical Bars | — |
| 03 | Remove Base Plate | — |
| 04 | Clean up | — |
| *Note: Steps 02-03 are interchangeable.* | | |

Table 22: Keysteps and Dependencies for Disassembling Camera Subassembly

| No. | Steps | Dependencies |
|---|---|---|
| 01 | Preparation | — |
| 02 | Remove Upper Horizontal Bar | — |
| 03 | Remove Middle Horizontal Bar | — |
| 04 | Remove Lower Horizontal Bar | — |
| 05 | Remove Tray Base | — |
| 06 | Repair Tray Base | 05 |
| 07 | Clean up | — |
| *Note: Steps 02-05 are interchangeable.* | | |

Table 23: Keysteps and Dependencies for Disassembling Trolley ITEM

| No. | Steps | Dependencies |
|---|---|---|
| 01 | Preparation | — |
| 02 | Remove Screws | — |
| 03 | Detach Tabletop | 03 |
| 04 | Clean up | — |

Table 24: Keysteps and Dependencies for Disassembling Circular Table

| No. | Steps | Dependencies |
|-----|-------|--------------|
| 01 | Preparation | — |
| 02 | Inspect and Clean | — |
| 03 | Attach Ram Chip | — |
| 04 | Attach Hard Disk | — |
| 05 | Attach Hard Disk Cables | — |
| 06 | Attach Side Plates | — |
| 07 | Attach Bottom Plate | 05 |
| 08 | Attach Top Plate | 03 |
| 09 | Attach Power Connection | 06 |
| 10 | Attach Connector Plates | 07 |
| 11 | Attach Wooden Plate | 10 |
| 12 | Attach Zip Ties | 11 |
| 13 | Plug in | 09 |
| 14 | Attach Monitor | — |
| 15 | Turn on | 13 |
| *Note: Steps 03-06 and 07-09 are interchangeable.* | | |

Table 25: Keysteps and Dependencies for Assembling Embedded Computing Unit

| No. | Steps | Dependencies |
|-----|-------|--------------|
| 01 | Preparation | — |
| 02 | Assemble Lower Hinge | — |
| 03 | Assemble Middle Hinge | — |
| 04 | Assemble Upper Hinge | — |
| 05 | Inspect | — |
| *Note: Steps 02-04 are interchangeable.* | | |

Table 26: Keysteps and Dependencies for Assembling IKEA Lamp

| No. | Steps | Dependencies |
|-----|-------|--------------|
| 01 | Preparation | — |
| 02 | Put on Gloves | — |
| 03 | Attach Feet | — |
| 04 | Connect Legs | — |
| 05 | Attach Upper Cover | 04 |
| 06 | Attach Lower Plate | 05 |
| 07 | Attach Crank | 05 |
| 08 | Adjust Height | 07 |
| 09 | Attach Tabletop | 08 |
| 10 | Clean up | — |
| *Note: For table without tabletop leave out step 09.* | | |

Table 27: Keysteps and Dependencies for Assembling Mechanical Table

| No. | Steps | Dependencies |
|---|---|---|
| 01 | Preparation | — |
| 02 | Put on Gloves | — |
| 03 | Connect Legs | — |
| 04 | Connect Center Bar | — |
| 05 | Assemble Upper Frame | — |
| 06 | Attach Hangers | — |
| 07 | Attach Tabletop | 05, 06 |
| 08 | Clean up | — |
| *Note: Steps 03-06 are interchangeable.* | | |

Table 28: Keysteps and Dependencies for Assembling IKEA Table

| No. | Steps | Dependencies |
|---|---|---|
| 01 | Preparation | — |
| 02 | Put on Gloves | — |
| 03 | Assemble Frame | — |
| 04 | Attach Horizontal Bar | 03 |
| 05 | Attach Wheels | 03 |
| 06 | Attach Plates | 04 |
| 07 | Attach Vertical Bar | 05 |
| 08 | Clean up | — |
| *Note: Steps 05-06 are interchangeable.* | | |

Table 29: Keysteps and Dependencies for Assembling Trolley Prototype

| No. | Steps | Dependencies |
|---|---|---|
| 01 | Preparation | — |
| 02 | Put on Gloves | — |
| 03 | Place CPU | — |
| 04 | Attach CPU Frame | 03 |
| 05 | Plug in Cables | — |
| 06 | Attach CMOS Battery | — |
| 07 | Attach Ram Chips | — |
| 08 | Attach Heat Sink | 04 |
| 09 | Attach Fan | — |
| 10 | Attach HDDs | — |
| 11 | Attach GPUs | 05 |
| 12 | Attach Chip Rack | — |
| 13 | Attach Duct | 08 |
| 14 | Close PC | — |
| 15 | Plug in and Test | 14 |
| 16 | Clean up | — |
| *Note: Steps 05-10 are interchangeable.* | | |

Table 30: Keysteps and Dependencies for Assembling Workstation Computer

| No. | Steps | Dependencies |
| --- | --- | --- |
| 01 | Preparation | — |
| 02 | Connect Base | — |
| 03 | Attach Vertical Bar | — |
| 04 | Attach Upper Bar | — |
| 05 | Attach Camera and Plate | — |
| 06 | Clean up | — |

*Note: Steps 02-05 are interchangeable.*

Table 31: Keysteps and Dependencies for Assembling Small Camera Assembly

| No. | Steps | Dependencies |
| --- | --- | --- |
| 01 | Preparation | — |
| 02 | Attach Base Plate | — |
| 03 | Attach Vertical Bars | — |
| 04 | Clean up | — |

*Note: Steps 02-03 are interchangeable.*

Table 32: Keysteps and Dependencies for Assembling Camera Subassembly

| No. | Steps | Dependencies |
| --- | --- | --- |
| 01 | Preparation | — |
| 02 | Attach Lower Horizontal Bar | — |
| 03 | Attach Middle Horizontal Bar | — |
| 04 | Attach Upper Horizontal Bar | — |
| 05 | Clean up | — |

*Note: Steps 02-04 are interchangeable.*

Table 33: Keysteps and Dependencies for Assembling Trolley ITEM

| No. | Steps | Dependencies |
| --- | --- | --- |
| 01 | Preparation | — |
| 02 | Attach Tabletop | — |
| 03 | Attach Screws | 03 |
| 04 | Clean up | — |

Table 34: Keysteps and Dependencies for Assembling Circular Table

| No. | Steps | Dependencies |
| --- | --- | --- |
| 01 | Inspect | — |
| 02 | Repair | 01 |
| 03 | Test | 02 |

*Note: Steps 01-03 can be repeated multiple times.*

Table 35: Keysteps and Dependencies for Repair Tasks

| No. | Steps | Dependencies |
| --- | --- | --- |
| 01 | Preparation | — |
| 02 | Disassemble | — |
| 03 | Inspect | — |
| 04 | Assemble | 02 |
| 05 | Test | 04 |
| 06 | Clean up | — |

*Note: Steps 02-05 can be repeated multiple times.*

Table 36: Keysteps and Dependencies for Open and Close Tasks

| No. | Steps | Dependencies |
| --- | --- | --- |
| 01 | Preparation | — |
| 02 | Perform Task | — |
| 03 | Clean up | — |

Table 37: Keysteps and Dependencies for Small Woodworking Tasks

| No. | Steps | Dependencies |
| --- | --- | --- |
| 01 | Preparation | — |
| 02 | Get Shipment | — |
| 03 | Transport | 02 |
| 04 | Unpack | 03 |

Table 40: Keysteps and Dependencies for Pickup Tasks

| No. | Steps | Dependencies |
| --- | --- | --- |
| 01 | Preparation | — |
| 02 | Find Devices | — |
| 03 | Load Trolley | 02 |
| 04 | Deliver and Unload Trolley | 03 |

*Note: Steps 02–04 can be repeated.*

Table 41: Keysteps and Dependencies for Inspection and Collection Tasks

| No. | Steps | Dependencies |
| --- | --- | --- |
| 01 | Preparation | — |
| 02 | Organize | — |
| 03 | Clean up | — |

Table 42: Keysteps and Dependencies for Organization Tasks

| No. | Steps | Dependencies |
| --- | --- | --- |
| 01 | Preparation | — |
| 02 | Disassemble Object | — |
| 03 | Perform Operation | 02 |
| 04 | Assemble Object | 03 |
| 05 | Clean up | — |

Table 38: Keysteps and Dependencies for Long Woodworking Tasks

| No. | Steps | Dependencies |
|---|---|---|
| 01 | Preparation | — |
| 02 | Prepare Shipment | — |
| 03 | Transport | 02 |
| 04 | Deliver | 03 |

Table 39: Keysteps and Dependencies for Delivery Tasks

## O  Limitations

While our dataset and paper aims to provide a comprehensive and multimodal benchmark for egocentric AI in industrial settings, we acknowledge its limitations.

**Domain specificity.** Our data is focused exclusively on industrial tasks, such as assembly, disassembly, cleaning, and tool manipulation in lab-like workshop environments. While this domain is highly relevant for practical applications of egocentric AI, the specificity limits generalisation to household, healthcare, or everyday scenarios. We believe, however, that this focus provides necessary depth and structure for studying procedural tasks, skill transfer, and collaborative work, which remain underexplored in the field.

**Participant diversity and scale.** The dataset comprises recordings from 20 participants (15 male, 5 female) with varied levels of industrial experience. Although this group reflects real-world labour demographics and includes meaningful variation in skill and behaviour, the sample size is relatively small. Future expansions should aim to include more participants, a greater balance across gender and age, and representation from different industrial domains (e.g., manufacturing, construction, maintenance).

**Limited tool and environment coverage.** The dataset includes a constrained set of tools and materials (e.g., clamps, drills, wooden components) recorded across a limited number of lab environments. As such, it may not fully capture the diversity of real-world factory floors, heavy machinery use, or mobile industrial contexts. Moreover, object taxonomies and spatial configurations are designed for structured tasks, which may not reflect more chaotic or unpredictable scenarios.

**Hardware and sensor limitations.** All egocentric data was captured using a specific wearable device (Meta Project Aria), which may not reflect the quality, field of view, or sensing capabilities of other platforms. Similarly, exocentric views are limited to static cameras with known occlusion cases. While our multimodal streams include RGB, depth, audio, SLAM, and eye gaze, synchronisation, processing and alignment errors may occasionally occur.

**Experimental design and baselines.** Some benchmark tasks, such as mistake detection or video QA, use relatively simple prompting or baseline models (e.g., zero-shot VLMs) due to computational constraints and the exploratory nature of this release. These baselines may not reflect the upper bound of achievable performance, and should be interpreted as reference points rather than definitive evaluations. Future work should explore fine-tuned models, multimodal fusion strategies, and stronger temporal reasoning baselines. Additionally, the local evaluations were conducted on the smallest VLM model (7B/8B) due to resource constraints.

**Annotation granularity.** While the dataset includes rich annotations (e.g., object interaction, speech, gaze, task steps), the granularity and consistency of some labels (particularly for complex or collaborative tasks) can vary across participants. Self-annotation was used in part to capture natural reasoning, but may introduce subjective interpretations or inconsistency. We proactively address this by reviewing the annotations. We encourage the community to build on this with more detailed or standardised annotation frameworks.

## P  Ethical Considerations and Broader Impact

Our dataset captures multimodal egocentric and exocentric recordings in industrial settings, where ethical considerations around privacy, consent, and the responsible use of data are especially important. Egocentric AI, by design, involves always-on perception that continuously captures not only the actions of the primary wearer but also their environment, speech, gaze behaviour, and potentially

other individuals present in the scene. This raises legitimate concerns about surveillance, data misuse, and personal autonomy.

In industrial contexts, these concerns are intensified by the structural relationship between employers and employees. The presence of wearable cameras and audio devices may be perceived as a form of monitoring, with implications for worker agency, psychological comfort, and trust. To mitigate these risks, all recordings in our dataset were conducted in controlled research environments with voluntary participation. Participants provided informed consent, were briefed on the scope of data capture (including eye gaze and speech), and had the option to review and remove any part of their data prior to submission. No covert or passive data collection was conducted. The dataset excludes sensitive personal identifiers, and we recommend that any downstream use of the data adhere to strict ethical guidelines and data minimisation principles.

At the same time, the broader impact of egocentric AI in industrial settings holds significant promise. Our dataset is designed not for surveillance, but to enable assistive and collaborative AI systems that can support real-world tasks such as tool-use guidance, procedural error detection, and hands-free contextual assistance. These applications have the potential to improve training for less experienced workers, enhance safety and efficiency, and provide real-time support in complex assembly or repair workflows. Moreover, our dataset includes annotations for skill variance, speech, and human-object interaction, allowing the community to study multimodal, human-centred AI from a diverse and realistic perspective.

We recognise that ethical considerations extend beyond the collection phase [67]. We encourage future work using this dataset to prioritise privacy-preserving techniques (e.g., on-device inference, frame redaction, speaker anonymization), and to avoid use cases that reinforce asymmetrical control or automated evaluation of worker performance without context. We also support the development of governance frameworks that involve workers in the deployment of such technologies. Our dataset aims to catalyse research that builds AI systems for collaboration, not control, where technology augments human skill, rather than replacing or monitoring it.

