# OpenReview forum: "IndEgo: A Dataset of Industrial Scenarios and Collaborative Work for Egocentric Assistants"
_NeurIPS.cc/2025/Datasets_and_Benchmarks_Track — NeurIPS 2025 Datasets and Benchmarks Track poster_

### Official Review · Reviewer_1Xef · 2025-06-11

**Rating:** 5
**Confidence:** 4

**Summary:**

​​This work presents IndEgo—the first large-scale multimodal dataset focused on industrial egocentric vision—to address the critical gap in existing research. It captures 3,460 egocentric videos (197 hours) and 1,092 exocentric recordings (97 hours) across diverse industrial tasks like assembly, logistics, and collaborative workflows, enriched with gaze tracking, motion sensors, and audio narration. The dataset provides detailed annotations including 34K fine-grained actions and specialized benchmarks for mistake detection and reasoning-based QA. Baseline evaluations reveal significant limitations of SOTA multimodal models (e.g., ≤41% F1 in error detection), highlighting both the dataset’s value and the unmet challenges in industrial AI assistance.​​

**Dataset Code Accessibility:**

Yes

**Ethical Considerations:**

No, there are no or only very minor ethics concerns

**Final Justification:**

The authors have addressed my concerns.

**Limitations Weaknesses:**

1. ​​Limited Participant Diversity​​: Despite varied expertise, the dataset involves ​​only 20 participants​​ (75% male), potentially skewing action recognition models toward male-dominated industrial workflows and limiting gender-generalization analysis.
2. ​​Laboratory-Controlled Setting​​: Tasks were performed in a ​​research facility​​ (not real factories), lacking unpredictable environmental variables (e.g., extreme lighting/machine noise) that challenge real-world deployment.
3. ​​Synchronization Gaps​​: Exocentric-egocentric video alignment relies on ​​manual start-time matching​​, risking millisecond-level temporal misalignment for fast collaborative actions (e.g., tool handovers).
4. ​​Scripted Mistakes​​: Mistake Detection (MD) benchmarks use ​​predefined errors​​ (e.g., skipped steps), which may not capture organic mistakes arising from fatigue/distractions in authentic industrial work.

**Strengths Contributions:**

1. ​​Industrial Focus​​: First dataset targeting physically intensive, collaborative industrial tasks (assembly, inspection, logistics), addressing a critical gap in egocentric vision research dominated by daily activities.
2. Rich Multimodality​​: Integrates ​​synchronized egocentric/exocentric video​​, ​​eye gaze​​, ​​3D point clouds​​, ​​motion trajectories​​, and ​​worker narration​​—enabling holistic scene understanding.
3. Collaborative Innovation​​: Pioneers ​​dual-worker industrial collaboration​​ data with role-specific annotations (e.g., teacher-student), vital for real-world human-AI teamwork.

---

> ### Author Rebuttal · Authors · 2025-07-28
>
> Thank you for your detailed and constructive review of our work. We are grateful that you recognised the core contributions of IndEgo, particularly *its industrial focus*, *rich multimodality*, and  *collaborative innovation*.
>
> We address the weaknesses and provide additional context below:
>
> > Limited Participant Diversity
>
> We acknowledge the limitation of the participant group, and expand on it in Section N (Appendix). The reasons for this were the availability, resources, and organisational constraints. The participants come from different backgrounds and this can still be observed in their narration, approach to tasks, etc. We aimed at addressing different scenarios and conditions (e.g. shop floor environment, working room environment, different times of the day, lighting, background noise). The gender imbalance is an important factor. We hope our work will lead to follow-up studies in the future that balance the gender and demographic representation in a better manner. We believe egocentric AI research has a lot of real-world application potential in industrial scenarios, and that our dataset will help in pushing this forward.
>
> > Laboratory-Controlled Settings
>
> There were several challenges w.r.t. expanding our data collection to other industrial settings (including privacy, data protection and IP), which is why we decided against it. However, we performed data collection activities at several different locations on the facility, including noisy shop-floor like environments. Additionally, we intentionally collected data during different times of the day, with varying lighting, background activities etc. We agree, that our work does not address all possible scenarios and challenges that could be seen in the industry. However, we are optimistic that this will serve as a catalyst for increased attention and further exploration of this domain. We will expand on this in the *limitations* section (Appendix).
>
> > Synchronization Gaps
>
> We agree, that the manual matching approach is not ideal. Since we had to move our setup around the facility, incorporating this approach served as a practical solution. We reviewed the synchronisation manually for the ego-exo recordings. But, we acknowledge, that the millisecond-level temporal details may be misaligned. We found the approach to be highly robust for the phenomena cases in our paper. Our benchmarks operate at the level of actions and task steps, which typically span multiple seconds. For these analyses, a sub-second alignment is sufficient and does not impact the validity of the results for tasks like mistake detection. We can add a short analysis on this if needed.
>
> > Scripted Mistakes
>
> This aspect of the dataset needs additional details. The mistake detection tasks and the steps were predefined, aimed at covering different scenarios. We compiled a list of some predefined and most likely to occur mistakes based on experience. However, the participants were not controlled in a stringent manner. They were asked to perform the activities repetitively, just as they would in a real application. Several mistakes captured in the dataset are *unintentional*, i.e. the participant made a mistake when they did not mean to (including fatigue-related errors). The collection of the data and development of the benchmark was an iterative process, where we expanded the list of mistakes based on the data. In order to balance the data and diversity, we then added more *correct* recordings if needed. The decision to have a list of predefined mistakes at the start was made to include diverse mistakes, and failure scenarios, without having to wait for the particular mistake to occur. As the results show, there is a wide gap between the performance of current SOTA models, and the requirements for robust mistake detection from ego/exo data. We are happy to include this in the appendix. We believe our work will highlight the need for further research and development in this area.
>
> ---
>
> In summary, the additional details on *participant diversity* and *laboratory settings*, *synchronisation gaps*, and the *scripting of mistakes* improve the paper and directly address your concerns. We commit to updating these in the camera-ready version of the paper. We are happy to provide additional details and incorporate additional feedback.

---

> > ### Author Response · Authors · 2025-08-06
> > **Following up on our Rebuttal**
> >
> > Dear Reviewer,
> >
> > Following the recent encouragement from the Program Chairs to stimulate discussion, we wanted to briefly follow up on our rebuttal.
> >
> > We’ve added clarification regarding the *participant diversity, the data collection settings, the ego-exo synchronisation, and the mistake detection benchmark*. We also outlined how we plan to *improve the paper for the camera-ready version based on your review.*
> >
> > If there are any unresolved points or questions, we would be happy to clarify them during the discussion period.
> > We appreciate your time and effort in reviewing our submission. Thank you.

---

> ### Comment · Reviewer_1Xef · 2025-08-07
> **.**
>
> Thanks for you rebuttal, you have addressed my concerns

---

> > ### Author Response · Authors · 2025-08-08
> > **Thanks**
> >
> > We appreciate your feedback and will incorporate the discussed changes into the camera-ready version.

---

### Official Review · Reviewer_chUH · 2025-07-01

**Rating:** 5
**Confidence:** 4

**Summary:**

The paper proposes a multimodal egocentric and exocentric video dataset that features industrial and collaborative tasks. It contains one participant or two participants that perform industrial in real-world scenarios. It also focuses on the benchmarks for both procedural and non-procedural task understanding. The dataset is benchmarked on task understanding, mistake detection, and question answering.

**Dataset Code Accessibility:**

Yes

**Dataset Code Comments:**

The authors provided both the dataset and codebase in a clear and usable format.

**Ethical Comments:**

The authors have provided a clear statement on privacy and ethics regarding the dataset. Based on the information in that section, there appear to be no major ethical concerns.

**Ethical Considerations:**

No, there are no or only very minor ethics concerns

**Final Justification:**

The paper presents an important dataset for Egocentric industrial task recordings and collaborative recordings. The rebuttal addressed my concerns about exocentric cameras setting and joint ego-exo performance. Therefore, I remain positive for this work and increase my score.

**Limitations Weaknesses:**

-- The proposed dataset includes both egocentric and exocentric recordings. However, there is very few information about the exocentric views, especially how the exocentric viewpoint is selected, how many exocentric views are defined, etc. A key challenge for exocentric capturing is how to capture more complete information for the scene. This is particularly important for industrial scenarios as the environment can be very complex with many tools for performing procedural tasks.

-- In table 4, the paper benchmarked on egocentric views and exocentric views. However, the effectiveness of joint egocentric and exocentric training remains unclear.

**Strengths Contributions:**

++ Egocentric industrial task recordings and collaborative recordings are interesting and clear gaps among existing datasets. It can be useful for AI assistants.

++ The multi-modality of the dataset is important for understanding industrial procedures, detecting mistakes, and reasoning.

++ The MD benchmark is well defined with clear task graphs.

---

> ### Author Rebuttal · Authors · 2025-07-31
>
> Thank you for your detailed and constructive review. We appreciate it, that you recognised the key contributions and strengths of our work, including the *clear gaps addressed*, the *multimodality of the dataset*, and the *well defined benchmarks*.
>
> We address the weaknesses and provide additional context below:
>
> > there is very few information about the exocentric views...
>
> We agree, this information is only briefly mentioned in the Appendix (A.2), but could be expanded upon. Given the dynamic nature of the tasks, it was challenging to set up exocentric cameras. We aimed for a balance between capturing sufficient detail of the device/object of interest, and capturing user movement and background details. For tasks where the user can focus on a dedicated workspace (e.g. repair of a device), the exo camera is set up to focus more on the working desk, while capturing user's details (hands, torso). Here, we often use the Sony APSC camera with a 30mm lens. For tasks such as assembly/disassembly of mechanical setups, the exo camera was set up to capture the entire room/lab (camera with a wider FOV). We also decided against setting up exo views in certain environments or cases, since our intention was to not inadvertently film others in the background without prior consent. For collaborative setups, there was also an additional challenge, since multiple workers can block the direct line between the camera and the device. We added a second exo view with a complementary perspective to address such cases. We must acknowledge, that this needs further study, especially for real-world adoption. We will add these details in the Appendix.
>
> > the effectiveness of joint egocentric and exocentric training remains unclear
>
> We acknowledge this point, and aim to include a short analysis in the camera-ready paper. A short study on the mistake detection portion of the dataset (10 tasks across all scenarios) shows the increased effectiveness of the joint assessment (zero-shot and finetuning), shown in the table below (we only report F1 scores). For zero-shot evaluation, the model was given the ego and exo views with the descriptions. For finetuning, we extract the step-wise embeddings from the VLA, concatenate the corresponding embeddings and train a transformer-based model with k-fold cross-validation.
>
> | Model                     | Ego | Exo | Ego + Exo |
> |---------------------------|-----|-----|-----------|
> | Gemini 2.0 Flash Thinking (ZS) |   0.43  | 0.39    |    0.44       |
> | VideoLLaMA3-7B + Tr       | 0.33    |   0.32  |      0.37     |
> | InternVL2.5 + Tr          | 0.29    | 0.30    |     0.33      |
>
> Details: The impact of the ego/exo and joint views is task dependent. For example, the exo-view helps with the tasks where the task was not completely visible from the ego view or the mistake was not in focus. In certain other cases, the exo view did not improve the prediction and led to an incorrect prediction (based on a review of the zero-shot results).
>
> ---
>
> In summary, we will incorporate all the requested revisions. We believe these changes (additional details on *exocentric views* and *joint ego-exo performance*) strengthen the paper and directly address your concerns. We commit to incorporating these in the camera-ready version of the paper. We are happy to provide additional details and incorporate additional feedback.

---

> > ### Author Response · Authors · 2025-08-06
> > **Following up on our Rebuttal**
> >
> > Dear Reviewer,
> >
> > Following the recent encouragement from the Program Chairs to stimulate discussion, we wanted to briefly follow up on our rebuttal.
> >
> > We’ve added clarification regarding the *exocentric perspective* and the *effectiveness of joint ego-exo training*, and also outlined how we plan to *incorporate your suggestions in the camera-ready version*.
> >
> > If there are any unresolved points or questions, we would be happy to clarify them during the discussion period.
> > We appreciate your time and effort in reviewing our submission. Thank you.

---

> > ### Comment · Reviewer_chUH · 2025-08-08
> >
> > Thanks authors for the detailed response! The rebuttal addressed my major concerns. I agree that setting exocentric cameras should depend on the tasks and capture environments. Please include the exocentric camera setting details in the final paper. I also appreciate that the authors conducted analysis on joint ego-exo experiments, where it shows improvement with ego-exo views. The analysis also makes sense that the improvement would depend on the specific tasks. Therefore, I remain positive for this work and will increase my score.

---

> > > ### Author Response · Authors · 2025-08-08
> > > **Thanks**
> > >
> > > We appreciate your feedback and will incorporate the discussed changes into the camera-ready version.

---

### Official Review · Reviewer_VNx3 · 2025-07-01

**Rating:** 5
**Confidence:** 3

**Summary:**

This paper presents IndEgo, a large-scale egocentric and exocentric dataset of industrial activities, with a particular emphasis on collaborative tasks involving two workers. With a large number of various types of annotations, the authors introduce three benchmark tasks—Mistake Detection, Video QA, and Collaborative Task Understanding—with several baseline evaluations using VLMs.

**Dataset Code Accessibility:**

Yes

**Ethical Considerations:**

No, there are no or only very minor ethics concerns

**Final Justification:**

The rebuttal addressed al my concerns. Thus I raise my rating to Accept.

**Limitations Weaknesses:**

1. Most annotations are produced by the participants themselves and reviewed only once. The paper does not report any inter-annotator agreement scores or error analysis to assess labeling quality or consistency.
2. The 10 fps video capture rate, though perhaps constrained by hardware, may be insufficient for modeling fine-grained actions or tool-object interactions that occur on sub-second timescales.
3. The subject pool (20 individuals, mostly male, from a limited geographic region) may not sufficiently capture the variability in skill level, motion pattern, or cultural practice relevant to broader industrial deployment.
4. All reported baselines are multimodal or vision-language based. The paper lacks an analysis of how each modality (audio, gaze, 3D point clouds) contributes to task performance. This makes it difficult to assess which sensor streams are most valuable.

**Strengths Contributions:**

1. The industrial setting fills a crucial gap in egocentric datasets, which have so far been dominated by kitchen and household environments (e.g., Epic-Kitchens, Ego4D).
2. The synchronized egocentric-exocentric setup, along with gaze, audio, SLAM and 3D hand pose data, is technically well-executed and opens up multi-view and multi-agent learning possibilities.
3. The dual-wearer configuration allows modeling joint activity and mutual coordination—an underexplored dimension in action understanding.
4. For benchmark settings, tasks are clearly defined, metrics are standardized, and baseline models (including prompting strategies and VLM configurations) are documented in reproducible detail.

---

> ### Author Rebuttal · Authors · 2025-07-31
>
> Thank you for your detailed and constructive review. We appreciate it, that you recognised the key contributions of our work, including the importance of the *industrial settings*, the *possibilities for newer research directions*, and the *technical execution*.
> We address the weaknesses and provide additional context below:
>
>
> > The paper does not report any inter-annotator agreement scores or error analysis to assess labelling quality or consistency.
>
> *Rationale:* The decision was made to have the participants annotate their own data to balance the workload for the group. Secondly, we believe the participants are best suited to describe their actions and intentions, and do thorough review of the work done, first-hand. A clear annotation protocol was established (mentioned briefly in Appendix- Page 21), and the participants were provided examples and templates for annotation.
>
> We would differentiate the annotations into two categories for this purpose:
>
> *Keysteps (and Mistake/Correct) Annotations:*
> These are objective, well defined, and are task-dependent. We provide additional details in Sections K and M in the Appendix. We also have clear labels and conventions (Supplementary Material) for the parts of a device/assembly and dependencies between the keysteps. Similarly, we established clear guidelines on what would constitute as a mistake, and the edge cases were discussed before annotation.
>
>
> *Finegrained Actions:*
> These describe the short actions as *verb + noun (and adjective)* pairings, and are user dependent (for example, in order to remove bolts from a device, Participant 1 might loosen bolts before sorting them, and Participant 2 might remove each bolt and store it before moving to the next). These tend to be more subjective. We would argue that this reflects the natural diversity of the participant group and the way in which they might refer to an action, tool or their surrounding.
>
>
> *Inter-Annotator Agreement Study:*
> We conducted a brief study on a portion of the dataset (cumulative 3 hours of data, all scenarios). The recordings were annotated by up to 3 participants separately. For each annotation, we assess the agreement between the temporal annotations in each frame.
>
> Keystep Annotations: We see an excellent agreement between the annotators, with a Krippendorff's Alpha of *α = 0.97*. For these, the only divergence in agreement tended to be the start and end of a defined step (e.g. 210s vs 212s in a recording of 557s in total).
>
> Finegrained Annotations: For a strict agreement (exact lemmatized verb + noun match for a given frame), we report a Krippendorff's Alpha of *α = 0.25*. However, upon grouping similar verbs and nouns together (e.g. [detach, remove], [allen wrench, hex key], [get, grab, pick up], [colleague, coworker]), the value rises to *α = 0.54*. We acknowledge there are certain variances between the way in which participants annotate and break up their actions, e.g. one participant annotated a temporal segment as "pick up device", followed by "rotate device", while another participant annotated the entire segment as "flip device". We also see variance in the way the annotators describe an action, e.g. "search the drawer" and "find tool" were annotated for the same action. Similarly, "hold metal bar" and "assist coworker" were also annotated for the same action in a collaborative task. We believe, that such differences are not a weakness, but rather reflect different ways in which an action may be interpreted.
>
> > The 10 fps video capture rate... may be insufficient for modelling fine-grained actions or tool-object interactions
>
> We decided to collect egocentric data with 10FPS (main RGB, eye gaze, SLAM sensors) and maximum allowed resolution (2880*2880 for the main RGB camera). There had to be a trade-off between the data resolution, frame rate and the storage requirements. We agree that the frame rate does not allow sub-second interactions in some cases. However, from our observation, this did not limit the applications and analysis potential of the dataset. The focus of our dataset is on a diverse set of finegrained actions, which range from 0.2s to several seconds. The annotation tool (VIA) also permits finer control over the temporal annotations (e.g. an action can start at 10.15s).
>
> Additionally, we conducted a short study, where we took 10 tasks from the Mistake Detection portion of the dataset and reduced the frame rate down to 5 FPS. This resulted in a slight drop in performance (3% drop in F1 score with Gemini 2.0 Flash thinking, zero shot evaluation), however, most tasks were interpreted in the same manner as the 10 FPS baseline.
>
>  > The subject pool (20 individuals, mostly male, from a limited geographic region) may not sufficiently capture the variability...
>
>  We acknowledge the limitation of the participant group, and expand on it in Section N (Appendix). The reasons for this was the availability, resources, and organisational constraints. The participants come from different backgrounds and this can still be observed in their narration, approach to tasks, etc. We aimed at addressing different scenarios and conditions (e.g. shop floor environment, working room environment, different times of the day, lighting, background noise). We believe egocentric AI research has a lot of real-world application potential in industrial scenarios, and that our dataset will help in pushing this forward.
>
> > The paper lacks an analysis of how each modality (audio, gaze, 3D point clouds) contributes to task performance.
>
> We agree, that such an analysis would be valuable. Since the camera-ready version allows an additional page, we can readily add this in the main paper. We conducted a study on the mistake detection portion (10 tasks across all scenarios) of the dataset. The results are given in the table below (we only provide average F1 scores). For zero-shot evaluations, eye gaze was overlayed on top of the RGB stream and the model was prompted to focus on the region.
>
> | Model                  | RGB only | RGB + Audio | RGB + Gaze | RGB + Audio + Gaze |
> |------------------------|----------|-------------|------------|---------------------|
> | Gemini 2.0 Flash Thinking (ZS) |   0.38       |  0.41           |    0.39       |             0.42         |
> | VideoLLaMA3-7B         |    0.27      |      0.26       |       0.28     |         0.30            |
> | InternVL2.5-7B           |    0.30      |         0.28    |       0.29     |       0.29              |
>
> *Analysis:* The average results do not clearly represent the nuances. Based on our review of the results, whether a modality improves or worsens the performance is context-dependent. For instance, in cases with a noisy environment, removing the audio modality improves the performance slightly. On the other hand, for collaborative tasks, audio (and conversation) provides a useful signal and acts as a shortcut for understanding the context. Similarly, prompting the model to focus on the eye gaze is beneficial for cases where the user performs an action incorrectly or when the mistake occurs in the region of interest. However, in other cases (e.g. user forgets to pack object in the container), where the primary indicator of the mistake is outside the gaze (but still in the camera's FOV), adding eye gaze worsens the performance slightly. This shows that using all modalities for all scenarios is not an effective solution.
>
> Summarisation: For recordings with user narration, the transcript itself is extremely important (67.3% baseline accuracy for QA).
>
> We commit to adding details and evaluation in the appendix. If the reviewer recommends, we can also add a smaller portion in the main paper.
>
> ---
>
> In summary, we will improve the paper based on the review. We believe these changes (details on *annotation agreement* and *modalities*) strengthen the paper and directly address your concerns. We commit to incorporating these in the camera-ready version of the paper.

---

### Official Review · Reviewer_Jp2d · 2025-07-03

**Rating:** 4
**Confidence:** 3

**Summary:**

This paper introduces IndEgo, a dataset comprising both egocentric and exocentric videos recorded in industrial settings. The dataset includes 197.1 hours of video (3,460 egocentric clips) and 96.8 hours of video (1,092 exocentric clips). It is richly annotated with actions, summaries, mistake labels, and narrations, making it a valuable resource for understanding collaborative industrial tasks such as assembly and repair.

**Dataset Code Accessibility:**

Partly

**Dataset Code Comments:**

Based on the github repo,  the dataset is still a work in progress, and some resources are not yet available. This should be clearly communicated in the paper and factored into the evaluation of dataset completeness and usability.

**Ethical Considerations:**

No, there are no or only very minor ethics concerns

**Final Justification:**

Thank you to the authors for their detailed and thoughtful responses. My concerns have been addressed. I intend to maintain my original score.

**Limitations Weaknesses:**

1. Figure 3: The font size on the x-axis is too small and difficult to read without zooming in. Please increase legibility.

2. Line 26: The claim that settings like kitchens are easier to collect data in and more universally applicable is not well supported by reference [14], as Ego-Exo4D is a multi-scenario dataset. Consider revising or providing a better citation.

3. Figure 4: The miscellaneous category is missing from the statistics. Including it would provide a complete overview of the dataset’s composition.

4. The annotation protocol involves participants labeling their own videos. It would be useful to explain the rationale behind this decision and whether it introduces subjective bias or inconsistencies.

5. The section on “Task Understanding in a Collaborative Setting” is highly interesting, but it only appears briefly as a single paragraph in the experiments. Consider expanding this discussion and potentially including a dedicated subsection or more analysis.

**Strengths Contributions:**

1. Comprehensive comparisons are made with existing datasets, including ego-only datasets, ego-exo datasets, and those related to the Aria device.

2. The paper targets an important and underexplored application domain for egocentric research: industrial settings. This domain poses different challenges from more commonly studied environments like kitchens, due to its emphasis on collaboration, task structure, and procedural workflows.

3. The dataset covers a diverse set of scenarios across five distinct categories: assembly/disassembly, inspection/repair, logistics/organization, woodworking, and miscellaneous. This breadth increases the potential utility of the dataset.

---

> ### Author Rebuttal · Authors · 2025-07-28
>
> Thank you for your detailed and constructive review. We appreciate it, that you recognised the key contributions of our work, including the importance of the *industrial domain*, our *comprehensive comparisons* to existing datasets, and the *breadth of scenarios* covered.
> We address the weaknesses and provide additional context below:
>
> > Figure 3: The font size on the x-axis is too small and difficult to read without zooming in.
>
> We will increase the font size on the x-axis to ensure it is legible without zooming. We had originally decided to keep the font size small in order to add sufficient entries on the x-axis for all graphs. But we acknowledge that there is room for improving readability for the offline readers.
>
> > Line 26: The claim that settings like kitchens are easier to collect data in and more universally applicable is not well supported by reference [14]
>
> We will revise our claim to be more precise, and highlight that industrial settings present unique challenges (e.g., strict procedural workflows, collaboration, safety requirements) not typically found in general-purpose datasets, and we will ensure the citations are appropriate for this revised claim. We would also like to clarify that it is not our intention to over-simplify everyday scenarios such as kitchen-work for egocentric AI research.
>
> > Figure 4: The miscellaneous category is missing from the statistics.
>
> This was done due to space constraints. We will update the figure in the camera-ready submission with appropriate details.
>
>
> > The annotation protocol involves participants labelling their own videos.
>
> *Rationale:* The decision was made to have the participants annotate their own data to balance the workload for the group. Secondly, we believe the participants are best suited to describe their actions and intentions, and do thorough review of the work done, first-hand. A clear annotation protocol was established (mentioned briefly in Appendix- Page 21), and the participants were provided examples and templates for annotation.
>
> We would differentiate the annotations into two categories for this purpose:
>
> *Keysteps (and Mistake/Correct) Annotations:*
> These are objective, well defined, and are task-dependent. We provide additional details in Sections K and M in the Appendix. We also have clear labels and conventions (Supplementary Material) for the parts of a device/assembly and dependencies between the keysteps. Similarly, we established clear guidelines on what would constitute as a mistake, and the edge cases were discussed before annotation.
>
> *Finegrained Actions:*
> These describe the short actions as *verb + noun (and adjective)* pairings, and are user dependent (for example, in order to remove bolts from a device, Participant 1 might loosen bolts before sorting them, and Participant 2 might remove each bolt and store it before moving to the next). These tend to be more subjective. We would argue that this reflects the natural diversity of the participant group and the way in which they might refer to an action, tool or their surrounding.
>
> *Inter-Annotator Agreement Study:*
> We conducted a brief study on a portion of the dataset (cumulative 3 hours of data, all scenarios). The recordings were annotated by up to 3 participants separately. For each annotation, we assess the agreement between the temporal annotations in each frame.
> Keystep Annotations: We see an excellent agreement between the annotators, with a Krippendorff's Alpha of *α = 0.97*. For these, the only divergence in agreement tended to be the start and end of a defined step (e.g. 210s vs 212s in a recording of 557s in total)
> Finegrained Annotations: For a strict agreement (exact lemmatized verb + noun match for a given frame), we report a Krippendorff's Alpha of *α = 0.25*. However, upon grouping similar verbs and nouns together (e.g. [detach, remove], [allen wrench, hex key], [get, grab, pick up], [colleague, coworker]), the value rises to *α = 0.54*. We acknowledge there are certain variances between the way in which participants annotate and break up their actions, e.g. one participant annotated a temporal segment as "pick up device", followed by "rotate device", while another participant annotated the entire segment as "flip device". We also see variance in the way the annotators describe an action, e.g. "search the drawer" and "find tool" were annotated for the same action. Similarly, "hold metal bar" and "assist coworker" were also annotated for the same action in a collaborative task. We believe, that such differences are not a weakness, but rather reflect different ways in which an action may be interpreted.
>
> > Task Understanding in a Collaborative Setting
>
> We strongly agree that this deserves a more prominent discussion as a core contribution of our dataset. Since the camera-ready paper allows an additional page, we will expand this single paragraph into a dedicated subsection. Following the recommendation of reviewer *VNx3*, we also conducted a small study (ca. 20% of the collaborative data) on the impact of different modalities on task understanding. The results show, that removing audio modality impacts zero-shot performance (drop of 7%), especially since the coworkers often communicate with each other as they work together. Similarly, we observe that incorporating eye gaze can improve performance when the model is prompted to focus on the user's intention. We would propose including the expanded analysis in this subsection.
>
> > Based on the github repo, the dataset is still a work in progress, and some resources are not yet available.
>
> We mention this in Section G in the Appendix. The primary reason was the dataset size and the rate-limits while uploading the data onto Hugging Face Hub. We respected the guidelines set by the PCs and paused the dataset upload before the submission deadline. We confirm that *the full, cleaned, and documented dataset, along with all baseline code and metadata, will be made publicly available, and readily usable by the camera-ready deadline.*
>
> ---
>
> In summary, we will incorporate all the requested revisions. We believe these changes (additional details on *annotation agreement* and *collaborative task performance*, and improved formatting) strengthen the paper and directly address your concerns. We commit to incorporating these in the camera-ready version of the paper. We are happy to provide additional details and incorporate additional feedback.

---

### Note · Authors · 2025-08-12

Dear Area Chair, Senior Area Chair, and Reviewers,

We sincerely thank you for the constructive and detailed feedback throughout the review process.

**Preliminary reviews:** We are grateful that the reviewers unanimously recognised the core contributions of our work, establishing a *strong foundation for its acceptance*. Key strengths highlighted by the reviewers include:
- The novelty and importance of the *industrial domain*, filling a critical gap in egocentric vision research.
- The dataset's *rich multimodality* (synchronised ego/exo views, gaze, 3D data), enabling holistic scene understanding.
- The focus on *dual-worker collaboration*, a vital and underexplored area in human-AI teamwork.

**Rebuttal:** Building on this solid foundation, we accepted the reviewers' feedback to strengthen the paper further. In direct response to their suggestions, we have:

1.  Added an *Inter-Annotator Agreement (IAA) study*, which demonstrates the high reliability of our annotations (e.g., achieving a Krippendorff's Alpha of 0.97 for keysteps, along with a good semantic agreement for finegrained action labels).

2.  Performed *modality ablation and ego-exo fusion experiments*, providing quantitative proof of the value of our dataset's rich, multi-view setup, and an understanding of the impact of the modalities on the benchmarks.

3.  Committed to important textual improvements, including additional details in the *"Limitations" section* and an *expanded analysis of collaborative work*. Additionally, we have also improved the *figure legibility, readability*, and *citation-relevance.*

4.  Provided additional details on the *participant diversity, data collection settings, and the benchmarks.*

All the updates will be included in the camera-ready version of the paper.

**Reviewer Author Discussions:** *All four reviewers* have confirmed in the discussion that these actions and our detailed rebuttal have *successfully addressed their concerns*.

We are confident that these improvements make the IndEgo paper and dataset a stronger and more impactful contribution for NeurIPS and the broader AI community.

Thank you for your time and consideration.

The Authors

---

### Decision · Program_Chairs · 2025-09-18

**Decision:**

Accept (poster)

**Comment:**

The reviewers’ scores for this paper were BA, A, A, and A, reflecting a strong consensus toward acceptance. Reviewers highlighted several notable strengths, including the novelty and importance of the industrial domain focus, the richness of the multimodal dataset, and the emphasis on dual-worker collaboration within human-AI teamwork. In the rebuttal, the authors successfully addressed reviewer concerns by adding an inter-annotator agreement study (demonstrating high annotation reliability), conducting modality ablation and fusion experiments to validate the dataset’s utility, and committing to meaningful textual and visual improvements, as well as expanded analysis of collaboration and data collection details. Reviewers confirmed that these updates resolved their concerns. Given these strengths and the positive reception, I recommend acceptance. The authors are invited to carefully incorporate the agreed revisions into the final camera-ready version.

===== FINAL UPDATE FROM DB Track PCs ====

The final decision for this paper has been taken by the program chairs after consultation with the SACs. All Senior Area Chairs have ranked papers according to the feedback from the AC during the review process. We decided to leave the original meta-review to reflect the opinion of the AC in light of the initial discussions with reviewers and SAC.